# Filming movies of attosecond charge migration in single molecules with high harmonic spectroscopy

Lixin He[1,2,5], Siqi Sun[1,5], Pengfei Lan [1,2] ✉, Yanqing He[1], Bincheng Wang[1], Pu Wang[1], Xiaosong Zhu[1,2], Liang Li[1,2], Wei Cao [1,2], Peixiang Lu [1,2,3] ✉ & C. D. Lin [4]

Electron migration in molecules is the progenitor of chemical reactions and biological functions after light-matter interaction. Following this ultrafast dynamics, however, has been an enduring endeavor. Here we demonstrate that, by using machine learning algorithm to analyze high-order harmonics generated by two-color laser pulses, we are able to retrieve the complex amplitudes and phases of harmonics of single fixed-in-space molecules. These complex dipoles enable us to construct movies of laser-driven electron migration after tunnel ionization of $N_2$ and $CO_2$ molecules at time steps of 50 attoseconds. Moreover, the angular dependence of the migration dynamics is fully resolved. By examining the movies, we observe that electron holes do not just migrate along the laser polarization direction, but may swirl around the atom centers. Our result establishes a general scheme for studying ultrafast electron dynamics in molecules, paving a way for further advance in tracing and controlling photochemical reactions by femtosecond lasers.

Understanding the ultrafast motion of electrons and nuclei after light-molecule interaction is fundamental to a broad range of chemical reactions and biological processes[1–7]. With the advent of attosecond light pulses at the dawn of the 21st century[8–10], various emerging techniques have been implemented by using attosecond pulse in combination with a femtosecond infrared (IR) pulse in a pump–probe configuration, such as attosecond extreme-ultraviolet (XUV) transient absorption spectroscopy[11,12], attosecond streaking technique[13] and mass/ion spectroscopy[14], making it possible to observe and partially manipulate the electronic degrees of freedom in molecular systems.

Quantum mechanically, after an electron was suddenly removed, the molecule is represented by a wave packet which is a complex-valued many-particle wavefunction of all the electrons and the nuclei of the molecule. Since electrons move on much faster (attosecond) timescales than the heavier nuclei (few to few-ten femtoseconds), within the first femtosecond, the nuclei positions can be assumed fixed

and the electron wave packet (EWP) can be expressed as a super-position of the electronic eigenstates, which will oscillate with the frequencies of the energy differences of the eigenstates. In previous publications by Cederbaum and coworkers[1,2], such time dependence of the electric charge density was called charge migration. This definition thus neglects the coupling from the escaping electron. On a longer timescale, the time-dependent EWP would cause the change of electronic potential landscape, forcing the nuclear wave packet to evolve in time. Ultrafast electron diffraction can be used to extract time-dependent nuclear dynamics[15–17], from which molecular movie can be constructed to provide a visualization of the nuclear motion. An equivalent attosecond electron movie of charge migration, however, is not readily available in conventional pump–probe experiments due to the limited time resolution by the pulse duration of the IR field[18,19]. Obtaining charge migration directly from experiment has been a challenging topic with great interest in the past decade.

[1]Wuhan National Laboratory for Optoelectronics and School of Physics, Huazhong University of Science and Technology, 430074 Wuhan, China. [2]Optical Valley Laboratory, 430074 Hubei, China. [3]CAS Center for Excellence in Ultra-intense Laser Science, 201800 Shanghai, China. [4]Department of Physics, Cardwell Hall, Kansas State University, Manhattan, KS 66506, USA. [5]These authors contributed equally: Lixin He, Siqi Sun. ✉e-mail: pengfeilan@hust.edu.cn; lupeixiang@hust.edu.cn

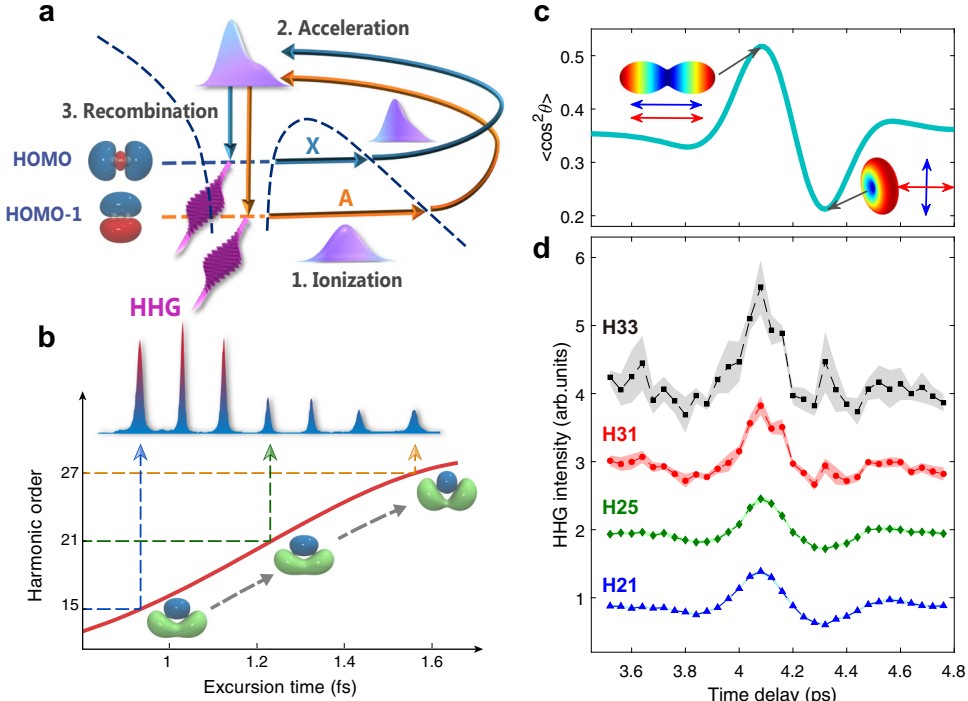

**Fig. 1 | Harmonic generation from N₂ molecule. a** HHG from each molecule can be described by the semi-classical three-step model: (1) ionization, (2) acceleration, and (3) recombination. After ionization, the molecular system is usually left in ground and several electronically excited states of the ion. In the HHG process, these ionic states provide different channels (Channels $X$ and $A$ for N₂) that connect the same initial and final states of the system. The interference of harmonics of these channels leads to the eventual harmonic emission. **b** Time-frequency mapping in the HHG process. For different high-harmonic orders, the freed electron spends different excursion times (the time interval between the ionization and recombination of the electron) in the continuum, providing a mapping between time and harmonic photon energy to enable temporal measurement of charge migration underlying the HHG process. Here we show only harmonics from the short-trajectory electrons due to its better phase matching in our experiment. **c** Time-dependent degrees of molecular alignment [$\langle\cos^2\theta\rangle(t)$] near the half rotational revival of N₂ in our experiment. The embedded false-color plots with prolate and oblate shapes correspond to molecular angular distributions at maximal alignment (4.08 ps) and maximal anti-alignment (4.32 ps) moments, respectively. Blue and red arrows indicate the directions of molecular alignment and the polarization of the driving laser pulse. **d** Measured HHG signals of H21, H25, H31, and H33 of N₂ at the time delays in (**c**). For clarity, the harmonic intensities here have been normalized to results measured for the isotropic case (i.e., without the alignment pulse). The results of H25, H31 and H33 have been shifted vertically. Shaded areas in (**d**) represent the standard deviation of the signals averaged over 1000 laser shots. The different spectral features of H31 and H33 from lower harmonics are due to contribution from the HOMO-1 orbital, especially at the time of maximal anti-alignment.

In ref. 19, a pseudo pump–probe protocol based on the recollision mechanism[20,21] of electrons in the laser field was discussed. In this process, the electron dynamics is triggered at the ionization step (pump), and probed at the recollision instant by the returning electron, either through elastic scattering with molecular ion or harmonic spectra by electron-ion recombination [see Fig. 1a]. Probing static or dynamic molecular structure using the diffraction image or high-order harmonic generation (HHG) are called laser-induced electron diffraction (LIED)[22–25] and high harmonic spectroscopy (HHS)[26–31], respectively. In HHS, the temporal resolution arises from the intrinsic frequency chirp of harmonic emission[32], i.e., different harmonic orders are associated with well-defined ionization-recombination delays spent by the electron in the continuum [see Fig. 1b], which can reach about 100 as for a commonly-used 800-nm laser field[26–28]. The spatial resolution of HHS comes from the de Broglie wavelength of the recombining electron and can reach sub-ångström[29–31].

Recently, using HHS, charge migration of ionized iodoacetylene (HCCI) was first reported by Kraus et al.[28]. Note that for charge migration derived from HHS the EWP evolves in the intense laser field while the original charge migration defined by Cederbaum et al.[1,2] are electron dynamics under the field-free condition. In this article, we make distinction by calling the electron dynamics in HHS laser-driven charge migration. As demonstrated in Ref. 28, to reconstruct electron dynamics from experimental harmonic spectra, one has to address many challenges. First, reconstruction is an inverse scattering problem, which is generally solved by iterative methods. Reconstruction is

possible only if the scattering theory is on firm ground. Second, the laser-driven electron dynamics is a property of a single molecule at a fixed position, thus it depends on the alignment angle of the molecular axis with respect to laser polarization. Since molecules cannot be aligned perfectly in experiment, the measured harmonics are the result of a coherent superposition of individual radiation weighted by the angular distribution of the molecules[33–35]. By taking angular averaged harmonic spectra from the experimental data, the phases of harmonics are severely compromised (see Supplementary Note 3), which would incur the loss of dynamics information of EWP that is at the heart of charge migration. Thus, it is critical to extract fixed-in-space single-molecule harmonic amplitudes and phases. Such a deconvolution or similar ones should be carried out in all measurements involving gas-phase molecules if single-molecule parameters are to be extracted, which, however, has not been properly treated in all of existing publications of HHS. In this work, this nontrivial deconvolution has been carried out. Besides, to extract charge migration, extra parameters, i.e., the population coefficients of different cation states, that describe the EWP of a single molecule should be obtained as well. Thus, it is also necessary to evaluate if adequate experimental data are available to ensure accurate retrieval of charge migration.

Here we demonstrate that, by properly addressing the above three issues, we have succeeded in filming movies of laser-driven charge migration in the molecular ion using HHS to reveal substantial sub-ångström electron migration. Since electron dynamics runs on the timescale of about 20 as (one atomic unit of time is 24 as), while the

emission time difference between two odd harmonics by an 800-nm laser is about 100 as, we shorten this interval by half, by generating both even and odd harmonics with a two-color laser field. With harmonic spectra from both single-color and two-color lasers in standard molecular alignment measurements, together with the powerful modern machine learning (ML) algorithm, we have abundant data to retrieve the complex dipole amplitude and phase of each fixed-in-space molecule to construct the charge migration movies. Thus, the alignment-angle-resolved molecular charge migration in the laser field has been fully characterized.

## Results

### Deciphering charge migration with HHS

We first explain how ultrafast charge migration is encoded in and how it can be identified from the molecular HHG signals. After strong-field tunneling ionization, the molecule in general is left in the ground as well as several nearby electronically excited states of the ion[1–4,36–38]. Once the electron is ejected, the occupation amplitudes of these electronic states would change with time, under the influence of the external laser field, creating a time-dependent many-electron wave packet, or equivalently, an electron–hole wave packet. The time dependence of the modulus square of the wave packet is called electron charge (or hole) migration. Such hole dynamics is encoded in the harmonic spectra of each single molecule[36–38].

To decipher electron dynamics from the harmonic spectra, we choose the two most widely studied molecules $N_2$ and $CO_2$ as the candidates to demonstrate our scheme. First, we consider $N_2$. As illustrated in Fig. 1a, the two channels (Channel $X$ and $A$) relevant to HHG in $N_2$ are associated with the ground $\tilde{X}^2\Sigma_g$ and first excited $\tilde{A}^2\Pi_u$ states of $N_2^+$ ion. In molecular orbital picture, these two channels correspond to ionization of an electron from the two highest-occupied molecular orbitals, HOMO and HOMO-1, respectively. Owing to the different symmetries of these orbitals, HHG from different ionization channels depends differently on molecular alignment. Figure 1c depicts the typical alignment distributions of $N_2$ molecules near the half revival period, while Fig. 1d shows the typical harmonic spectra measured in the experiment (for experimental details, see Methods and Supplementary Note 1). When molecules are aligned (anti-aligned) with respect to laser polarization, the signals of lower order harmonics (e.g., H21 and H25) are strong (weak) because they are generated from the HOMO orbital. For higher orders, like H31 and H33, the signals receive a substantial contribution from the HOMO-1 orbital and thus becomes quite large when the molecules are anti-aligned. These properties of the harmonic spectra of $N_2$ have been well-studied experimentally[39] and can be calculated with quantitative rescattering (QRS) theory[40–43]. Similar results are also observed in HHG from $CO_2$ molecule (see Supplementary Note 2). The different alignment dependence of the contribution to HHG from multiple molecular orbitals allows us to access their complex population amplitudes in the complex wave packet of the molecular ion, from which ultrafast charge migration can be uncovered.

### Retrieval of single-molecule dipole

For a parallel configuration of the polarizations of the alignment and probe pulses, the time-dependent harmonic signal is given by[33,34]

$$S(\omega, \tau) = \left| \int_0^\pi D(\omega, \theta)\rho(\theta, \tau) \sin\theta \, d\theta \right|^2, \qquad (1)$$

where $\theta$ is the alignment angle of the molecule, $\tau$ is the time delay between these two pulses, $D(\omega, \theta)$ is the fixed-angle-dependent total dipole moment from the single-molecule response, and $\rho(\theta, \tau)$ is the angular distribution of molecules at delay $\tau$. In our experiment, $\rho(\theta, \tau)$ is determined with method in Ref. 44. Note that perfect molecular alignment cannot be achieved in experiment. Our simulations show

that single-molecule harmonic amplitudes and phases deviate substantially from the angle-averaged ones (see Supplementary Note 3).

To obtain molecular-frame electron dynamics, the first and foremost step (omitted in all previous literature) is to decode angle-dependent single-molecule dipole moment $D(\omega, \theta)$ (including both amplitude and phase) from the measured harmonic signals. However, to solve this inverse problem directly is rather difficult since Eq. (1) is nonlinear and is ill-posed. Here, we introduce ML to this problem. The modern ML algorithm has been demonstrated to have remarkable abilities in characterizing complex sets of data with a high degree of accuracy, and has been widely utilized in genetics[45], condensed-matter physics[46], and material science[47]. We show here that it can effectively deal with the complicated decoding in HHS (see Methods and Supplementary Note 4).

The second step of our reconstruction is to disentangle the multichannel contributions from the total single-molecule dipole moment $D(\omega, \theta)$ obtained in the first step. Driven by a one-color 800-nm probe pulse, only one set of $D(\omega, \theta)$ can be obtained for each harmonic order which is insufficient for the reconstruction. To overcome this problem, we employed an additional parallel two-color laser field to generate harmonics. This two-color field consists of an intense 800-nm fundamental field and a weak second-harmonic (SH) field. The SH field is weak ($\sim 2 \times 10^{-3}$ of the fundamental field), which barely alters the electron dynamics of the molecular ion (see Supplementary Note 5), but could affect the intensities and phases of even harmonics as in ref. 48. Thus, measurement of harmonics at different relative phases between the fundamental and SH fields can replenish the additional data set needed to retrieve multichannel contributions in the second step. Figure 2a, b show the time-dependent signals of H22 and H27 from $N_2$ measured as a function of the relative phase of the two-color laser fields. One can see that the modulation of HHG intensity depends sensitively on the relative phase, and different harmonic orders present different dependences. Applying the ML-based reconstruction procedure to each harmonic order, we can obtain the time-dependent complex mixing coefficients of the multiple orbitals of the molecular cation, for each fixed-in-space angle, at the excursion time when the recombination of that harmonic order occurs, as illustrated in Fig. 1b.

Figure 2c shows the reconstructed population amplitudes of the ground $\tilde{X}$ state of the $N_2^+$ ion for four specific alignment angles versus the excursion time. The circles indicate the data extracted from the experiment for harmonics from H15 to H27 (including even and odd orders). Figure 2d shows the reconstructed relative phase between the wavefunctions of $\tilde{X}$ and $\tilde{A}$ states. With these parameters, a complex-valued time-dependent wave packet from the two holes $\tilde{X}$ and $\tilde{A}$ can be constructed for $N_2^+$.

We have also carried out the measurements for $CO_2$ molecules. Harmonic spectra of $CO_2$ involves three electronic states of the ion, the ground $\tilde{X}$ (HOMO), the first $\tilde{A}$ (HOMO-1) and second $\tilde{B}$ (HOMO-2) excited states[36,37]. Using the same procedure as for $N_2$, the reconstructed populations and relative phases of these three ion states are shown in Fig. 2e, f and g, h, respectively. With these parameters, a complex-valued time-dependent wave packet from the three holes $\tilde{X}$, $\tilde{A}$ and $\tilde{B}$ can be constructed for $CO_2^+$. To evaluate our reconstructions, we have carried out calculations based on the time-dependent density functional theory (TDDFT) to simulate the parameters we obtained. The theoretical calculations are in reasonable agreement with the reconstructions (see Supplementary Note 5).

### Filming attosecond charge migration in the laser field

To visualize hole dynamics, we have calculated the modulus square of the wave packet versus time for $N_2^+$ and $CO_2^+$ with the data in Fig. 2c–h. To quantify attosecond charge migration processes, we first examine hole dynamics of $CO_2^+$ when the fixed-in-space molecule is aligned parallel to the laser polarization axis. The results are shown in Fig. 3a, b. To help understanding, we plot the relevant molecular orbitals in

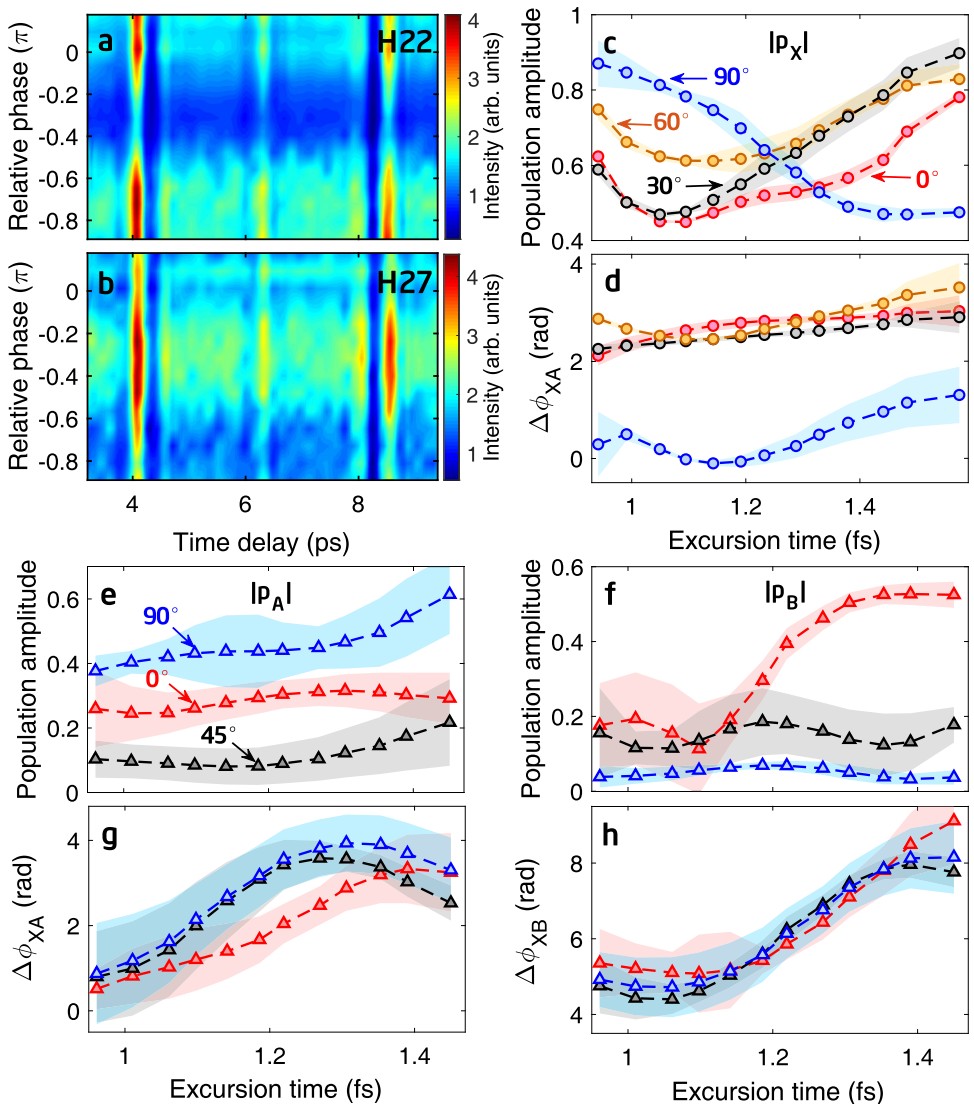

**Fig. 2 | Two-color HHS scheme for probing molecular charge migration in N₂ and CO₂. a, b** Measured harmonic signals of H22 and H27 in the two-color experiment of N₂. Measurements at different relative phases of the two-color lasers provide sufficient data sets for resolving the multichannel electronic dynamics. **c, d** Reconstructed population amplitude of the $\tilde{X}$ state ($|p_X|$) and the relative phase between the wavefunctions of $\tilde{X}$ and $\tilde{A}$ states ($\Delta\phi_{XA}$) in N₂ for the alignment angles of 0° (red circles), 30° (black circles), 60° (orange circles), and 90° (blue circles), respectively. Each circle represents the excursion time of a single harmonic order,

as depicted in Fig. 1b. **e, f** Reconstructed population amplitudes of the $\tilde{A}$ state ($|p_A|$) and $\tilde{B}$ state ($|p_B|$) in CO₂ for the alignment angles of 0° (red triangles), 45° (black triangles), and 90° (blue triangles), respectively. **g, h** Reconstructed relative phases between the wavefunctions of $\tilde{X}$ and $\tilde{A}$ ($\Delta\phi_{XA}$) and $\tilde{X}$ and $\tilde{B}$ ($\Delta\phi_{XB}$) states in CO₂ for the above three angles. Shaded areas in (**c**–**h**) represent the standard deviations of the reconstructions. These amplitudes and phases retrieved for each harmonic order are used to construct the wave packet of the hole dynamics of the cations.

Fig. 3c. Figure 3a presents hole densities extracted at the recombination times of harmonics H15 to H26. Here, $t = 0$ is when electron is born. In the figures, the frames are separated by steps of about 50 as. At a quick glance, we can see substantial hole migration from $+x$ side to the $-x$ side over the 0.5 fs duration. To take a closer look at how the hole migrates, in Fig. 3b we zoom in the region of $x = [-1.2, 1.2]$ a.u. In the first 0.25 fs, we can see that the hole density coalesces in the middle region along the $x$-axis slowly from the $+x$ side to the $-x$ side, but in the second half, the pace turns faster where at later time we can see some density is moving back to the $+x$ side. In fact, its time evolution is more like a counterclockwise swirling than a direct migration. To observe the continuous evolution of hole density, we have constructed movies at steps of 10 as by interpolating the mixing coefficients in the complex wave packets shown in Fig. 2e–h (see Supplementary movie 1).

Since quantum mechanics can predict single-hole behavior only statistically, it is inappropriate to associate such movement with

trajectory or speed. It is more proper to think such movement like that of a flock of migrating birds. Each bird has its own mind and there is nothing to govern the trajectory of each bird, yet the flock will move on. In the presence of an external disturbance, the movement of the flock will change. So is the migration of an electron or a hole in the presence of the laser field. To describe the movement of charge density of a quantum system, consider the equation of continuity, $\frac{\partial \rho_e}{\partial t} + \nabla \cdot \mathbf{J} = 0$, where $\rho_e$ is the probability density and $\mathbf{J}$ is the probability current density. By calculating them from the complex hole wave packet, we can obtain the total flux crossing the $x = 0$ plane at each time instant, as shown by the red line in Fig. 3d. We have confirmed that the total flux change is equal to the rate of change of the total charge density. Likewise, the total flux crossing the $y = 0$ plane can be calculated (the blue line). The total flux varies rapidly versus time, demonstrating attosecond dynamics.

Our reconstruction method for each harmonic obtains the occupation amplitude and phase for all angles of each fixed-in-space

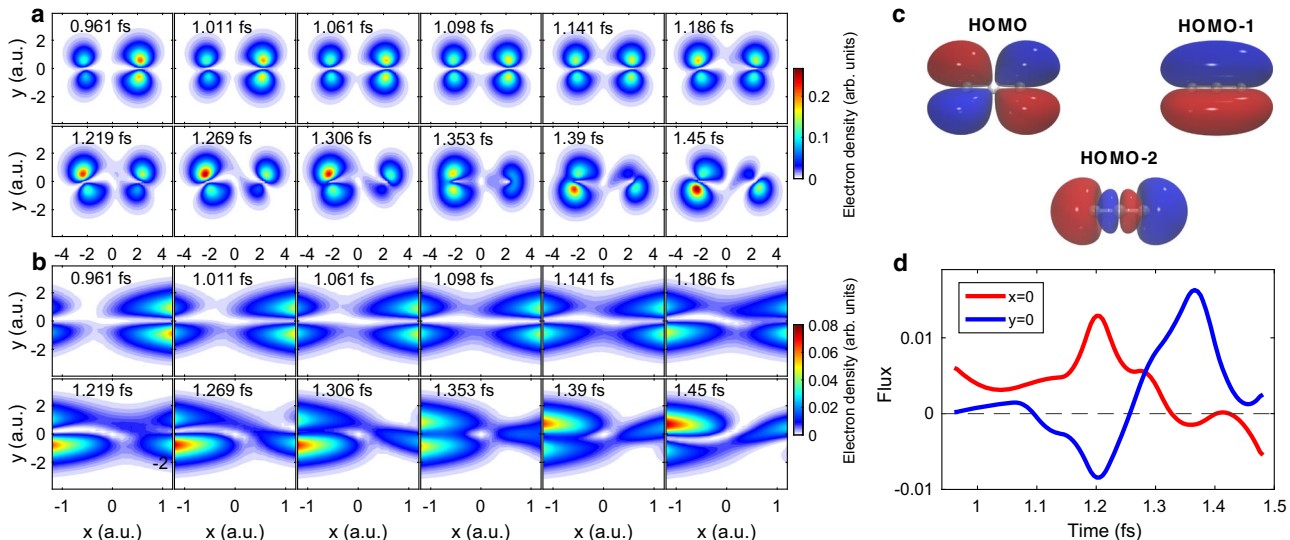

**Fig. 3 | Reconstruction of attosecond charge migration in $CO_2^+$. a** Snapshots of the reconstructed hole densities for the alignment angle of 0°. **b** Enlarged views of the results in (**a**) in the range from $x = -1.2$ a.u. to $x = 1.2$ a.u. **c** HOMO, HOMO-1, and HOMO-2 orbitals of $CO_2$ molecule. **d** Flux of charge density crossing the $x = 0$ (red) and $y = 0$ (blue) planes. Positive value means charge migration from $+x$ ($+y$) side to $-x$ ($-y$) side.

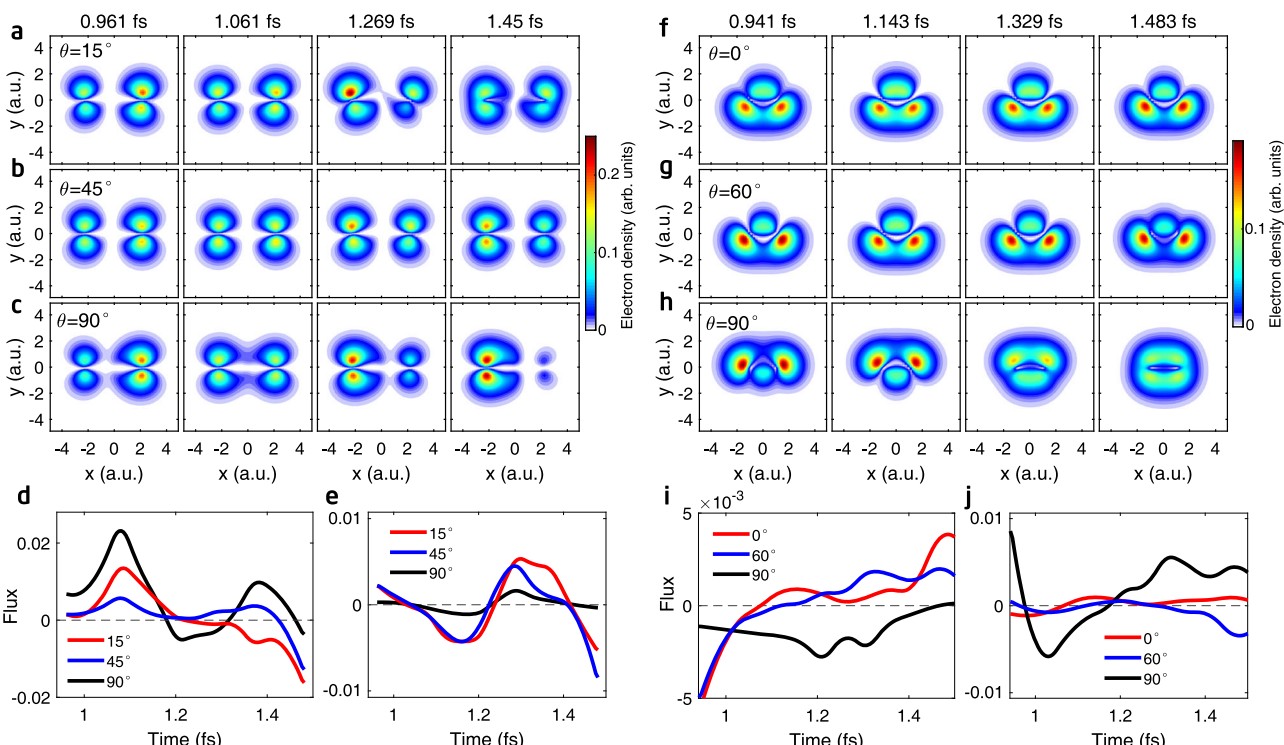

**Fig. 4 | Alignment-angle-dependent attosecond charge migration in $N_2^+$ and $CO_2^+$. a–c** Snapshots of the reconstructed hole densities in $CO_2^+$ for the alignment angles of 15°, 45° and 90°, respectively. **d** Flux of charge density crossing the $x = 0$ plane for the three angles in (**a–c**). **e** Same as (**d**), but for the $y = 0$ plane. Positive value means charge migration from $+x$ ($+y$) side to $-x$ ($-y$) side. **f–h** Snapshots of the reconstructed hole densities in $N_2^+$ for the alignment angles of 0°, 60°, and 90°, respectively. **i** Flux of charge density crossing the plane of $x = -1.5$ a.u. for the three angles in (**f–h**). **j** Same as (**i**), but for the $y = 0$ plane.

molecules. In Fig. 4a–c, we show a sample of hole density plots of $CO_2$ molecule for alignment angles of 15°, 45°, and 90°, respectively. For the same 0.5 fs duration, we can clearly see that the hole density does undergo significant change and vary significantly with alignment angles. These results prove that one should not obtain single-molecule harmonic phase and amplitude directly from the experiment without angular deconvolution. In Fig. 4d, e we also calculated the total flux across the $x = 0$ and $y = 0$ planes, respectively, for the three angles. In

Supplementary Movies 2, 3, 4 we show the movies of hole migration for these three angles.

In Fig. 4f–h, we show the reconstructed hole densities in $N_2$ at several selected excursion times for the alignment angles of 0°, 60°, and 90°, respectively. The hole densities are symmetric about the $x = 0$ plane for all alignment angles because of the symmetry of the HOMO and HOMO-1 orbitals of $N_2$. At 0° and 60°, only small migration can be seen, because HHG from HOMO-1 is weak. At 90°, HOMO-1 becomes

important and one can see faster change of charge density at later excursion time. These results are consistent with the time dependence of total flux across the $x = -1.5$ a.u. and $y = 0$ planes, respectively, as shown in Fig. 4i, j. Note that the hole density is symmetric about the $x = 0$ plane, thus the flux crossing the $x = 0$ plane is very small. Hole migration for $N_2$ has also been constructed (see Supplementary Movies 5–8). In the constructed movies, the time span is associated with the excursion time of the harmonic orders (H15-H26 for $CO_2$, and H15-H27 for $N_2$) used in the reconstruction, which is limited to approximately 0.94 fs to 1.5 fs for an 800-nm driving laser as illustrated in Fig. 1b. This time window can be further expanded by using a long wavelength driving laser[28]. The extended time window of the molecular movie will help to capture the effect of nuclear motions on the electron dynamics at longer time.

Comparing to results previously reported by Kraus et al.[28], we are able to make movies of laser-driven charge migration at steps of 10 as by interpolating the experimental data, for molecules at all alignment angles. This is because we were able to retrieve single-molecule complex dipole at each fixed-in-space angle and probe charge migration at recombination times for both even and odd harmonics at every 50 attoseconds, instead of 100 attoseconds if only odd harmonics are measured. With twice as many data points within the same interval, it is more accurate to interpolate. Figure 3b has shown that such 10 as resolution is needed in order to follow the charge migration in $CO_2$. Before closing, we mention that we have confirmed that charge migration dynamics of single molecules constructed directly from experimental harmonic spectra would cause large errors (see Supplementary Note 6), and that charge migration under field-free condition evolves slower than when electrons are under the intense laser field (See Supplementary Note 7). The latter is similar to the familiar field-free two-level oscillation is slower than the Rabi oscillation when the two levels are driven by an intense laser pulse. In experiment, the influence of the strong laser field on the charge migration process may be reduced by using a few-cycle fundamental pulse instead of a multi-cycle pulse. But in the few-cycle regime, it may increase the difficulty to build the time-frequency mapping due to the less sharp spectrum structure of each individual harmonic.

## Discussion

In summary, we have used a ML-based HHS method to construct movies of laser-driven charge migration in a molecular ion at its natural timescale of 10 as to follow the change of hole density during the time between electron is removed till it has recombined with the ion to emit harmonics. We presented the migration dynamics at the most fundamental level for each single fixed-in-space molecule at all alignment angles from the experimental HHG spectra. The method presented here in principle can be extended to other molecules, but in reality, it will be limited by the number of molecular orbitals in the wave packet. As the number of mixing parameters in the wave packet increases, larger amount of experimental data would be needed for retrieval. Looking ahead, the method may be extended to other molecules where harmonic spectra have been well studied to identify other possible migration pathways, for examples, in ring molecules. Finally, we want to emphasize that the method of retrieving single-molecule parameters used in the present work should be extended to other ultrafast experiments, including tomography, time delays, photoelectron momentum spectra, just name a few, for any gas-phase targets where the experimental data are always resulting from coherent or incoherent superposition of signals from individual single fixed-in-space molecules weighted by the angular distribution of the molecules.

## Methods
### Experimental methods
Our experiment is carried out by using a commercial Ti:sapphire laser system (Legend Elite-Duo, Coherent, Inc.), which delivers 35-fs,

800-nm laser pulses at a repetition rate of 1 kHz. The output laser is split into an alignment and a probe pulse. The alignment pulse with moderate intensity is used to induce nonadiabatic alignment of molecules along its polarization. The intense probe pulse has been used either directly (one-color scheme) or remolded to a parallel two-color laser field (see Supplementary Note 1) to interact with the aligned molecules to generate high-order harmonics. The alignment and probe pulses are parallel in the polarization. A motorized delay line is installed in the arm of the alignment pulse to adjust the time delay between the two pulses. These two pulses are focused to a supersonic gas jet by a spherical mirror ($f = 250$ mm). The gas jet is placed 2 mm after the laser focus to ensure good phase matching for short-trajectory harmonics, and the backing pressure is maintained at 0.8 bar. The generated high harmonics are detected by a homemade flat-field soft x-ray spectrometer. In the one-color experiment, the harmonic signals measured at different time delays [Fig. 1d and Supplementary Fig. 2e] are used to identify the multiple orbitals effect in HHG. In the two-color experiment, the time-dependent harmonic signals are measured at various relative phases of the two-color laser field, providing a two-dimensional data set for decomposing the multiple orbital contributions in HHG process.

### Reconstruction methods
Our reconstruction procedure has two steps. The first step is to retrieve single-molecule dipole moment from the measured time-dependent HHG signals. HHG from aligned molecular ensemble is expressed as Eq. (1). For a given harmonic order, Eq. (1) can be expanded as[49,50]

$$S(\tau) = \left[\int_0^\pi D(\theta)\rho(\theta,\tau)\sin\theta\,d\theta\right]^* \left[\int_0^\pi D(\theta)\rho(\theta,\tau)\sin\theta\,d\theta\right]$$
$$= \int_0^\pi \int_0^\pi D^*(\theta_1)D(\theta_2)\rho(\theta_1,\tau)\rho(\theta_2,\tau)\sin\theta_1\sin\theta_2\,d\theta_1\,d\theta_2. \quad (2)$$

Let

$$R(\theta_1,\theta_2) = \mathrm{Re}[D^*(\theta_1)D(\theta_2)], \quad (3)$$

$$\rho(\theta_1,\theta_2,\tau) = \rho(\theta_1,\tau)\rho(\theta_2,\tau)\sin\theta_1\sin\theta_2, \quad (4)$$

and discretize $\theta_1, \theta_2$ in the range of $[0, \pi]$ with a step of 0.01 rad, the $S(\tau)$ then can be rewritten as

$$S(\tau) = \sum_p \sum_q R(\theta_1^p, \theta_2^q)\rho(\theta_1^p, \theta_2^q, \tau)\,d\theta_1\,d\theta_2. \quad (5)$$

Note that in Eq. (3), the imaginary part of $D^*(\theta_1)D(\theta_2)$ is omitted due to its asymmetry upon the exchange of $\theta_1$ and $\theta_2$, which will vanish after the convolution. In Eqs. (2) and (4), the molecular axis distribution $\rho(\theta,\tau)$ and therefore $\rho(\theta_1,\theta_2,\tau)$ can be determined from the one-color experiment with the method in[44]. To retrieve single-molecule dipole moment $D(\theta)$, we first solve the matrix $R(\theta_1,\theta_2)$ according to Eq. (5). Here, we utilize a widely-used ML algorithm, sparse representation[51], to do the retrieval. In the reconstruction, we first build a dictionary matrix for $S(\tau)$ by expanding the $R$ matrix with a series of two-dimensional Legendre polynomial basis functions. The expansion coefficients then can be solved with the LASSO regression[52] by minimizing the $l_1$-norm of the coefficient vector and also the difference between the reproductions and the measurements. With $R(\theta_1,\theta_2)$ retrieved, we can obtain the amplitude and phase of the dipole moment $D(\theta)$ in terms of the derivations of Eq. (3). The validity and robustness of this algorithm have been validated by simulation experiments (see Supplementary Note 4). Applying this procedure to measurements at different relative phases $\alpha$ of the two-color laser field, we can get a set of $D(\theta; \alpha)$ for each harmonic order.

The second step of the reconstruction is to decompose the multichannel contributions from the total dipole moment $D(\theta; \alpha)$ obtained in the first step. In molecular HHG, the total dipole moment $D(\omega, \theta)$ is a coherent superposition of the induced dipole moment of each emission channel, i.e.,

$$D(\omega, \theta) = \sum_i D_i(\omega, \theta), \qquad (6)$$

where $D_i(\omega, \theta)$ is the induced dipole moment of each emission channel. The subscript $i$ denotes the involved molecular orbitals (molecular ion states) during the HHG process. Consider the initially populated molecular ion state $i$, $D_i(\omega, \theta)$ is given by

$$D_i(\omega, \theta) = d_{ion}^i(\omega, \theta) a_{acc}^i(\omega) d_{rec}^i(\omega, \theta), \qquad (7)$$

where $d_{ion}^i(\omega, \theta)$ and $d_{rec}^i(\omega, \theta)$ are the transition dipole moments related to the ionization and recombination steps, respectively. $a_{acc}^i(\omega)$ denotes the propagation amplitude of the EWP in the continuum. $d_{ion}^i(\omega, \theta)$ can be expressed as

$$d_{ion}^i = \sqrt{\eta_i} \langle \Psi_0(N) | \tilde{D} | \Phi_i(N-1)\chi_k \rangle \qquad (8a)$$

$$= \sqrt{\eta_i} \langle \psi_i | r | \chi_k \rangle. \qquad (8b)$$

Here $\eta_i$ is the ionization rate, $\Psi_0(N)$ is the ground state of the N-electron molecule, $\Phi_i(N-1)$ is the ground state or excited state of the (N-1)-electron molecular ion, where a molecular orbital $\psi_i$ has been removed to a continuum state $\chi_k$. Both many-electron wavefunctions $\Psi_0(N)$ and $\Phi_i(N-1)\chi_k$ are properly antisymmetrized and $\tilde{D}$ is the dipole operator from all the electrons. In obtaining Eq. (8b), we assume that all the molecular orbitals are in the neutral and the ion does not change before and after ionization.

To calculate the recombination transition dipole, we take into account that the molecular ion has been modified by the laser field during the time interval between ionization and recombination to $\Phi_i'(N-1)$, where

$$\Phi_i'(N-1) = \sum_j C_{ij}(\theta)\Phi_j(N-1). \qquad (9)$$

Thus, the recombination dipole

$$\begin{aligned} d_{rec}^i &= \langle \Phi_i'(N-1)\chi_k | \tilde{D} | \Psi_0(N) \rangle \\ &= \sum_j C_{ij}(\theta) \langle \chi_k | r | \psi_j \rangle. \end{aligned} \qquad (10)$$

Inserting Eq. (10) to Eq. (7), we obtain

$$D_i(\omega, \theta) = \sum_j C_{ij}(\theta) d_{ion}^i(\omega, \theta) a_{acc}^i(\omega) d_{rec}^j(\omega, \theta). \qquad (11)$$

Eq. (11) implies that hole hopping occurs before recombination. The degree of hopping depends on laser parameter and the alignment angle $\theta$ of the molecule. With Eq. (11), the total dipole moment for HHG can be expressed as

$$D(\omega, \theta) = \sum_{ij} C_{ij}(\theta) \bar{D}_{ij}(\omega, \theta), \qquad (12)$$

where

$$\bar{D}_{ij}(\omega, \theta) = d_{ion}^i(\omega, \theta) a_{acc}^i(\omega) d_{rec}^j(\omega, \theta). \qquad (13)$$

In our reconstruction, only the most relevant molecular orbitals $\psi_j$ are considered (that is, $\tilde{X}$ and $\tilde{A}$ states for $N_2$, and $\tilde{X}$, $\tilde{A}$, and $\tilde{B}$ states for $CO_2$). The dipole moment $\bar{D}_{ij}(\omega, \theta)$ is calculated under the experimental laser conditions. The complex-valued coefficients $C_{ij}(\theta)$ are

directly associated with the electron dynamics in the molecular ion. To determine $C_{ij}(\theta)$, we have performed two-color experiment. Since the SH field in the two-color experiment is weak enough and hardly alters the laser-driven electron dynamics, the coefficients $C_{ij}(\theta)$ can be assumed to be independent of the relative phase $\alpha$. The coefficients $C_{ij}(\theta)$ are then retrieved from the dipole moment $D(\theta; \alpha)$ obtained at different relative phases $\alpha$ by solving Eq. (12) with the genetic algorithm.

With the coefficients $C_{ij}(\theta)$ retrieved, we can further calculate the wavefunction of the molecular ion at the recombination instant as

$$\psi_{sum} = \sum_i \gamma_i(\theta)\psi_i' = \sum_i \sum_j \gamma_i(\theta) C_{ij}(\theta)\psi_j, \qquad (14)$$

and the population coefficient of the molecular orbital $\psi_j$ as[28]

$$p_j = \sum_i C_{ij}(\theta)\gamma_i(\theta), \qquad (15)$$

where $\gamma_i(\theta)$ is the initial population of the molecular ion state $i$, which is related to the alignment-angle-dependent ionization rate $\eta_i(\theta)$ by $|\gamma_i(\theta)|^2 = \eta_i(\theta)/\Sigma_i\eta_i(\theta)$. In our work, the alignment-angle-dependent ionization rates of each molecular orbital are simulated by the MO-ADK theory[53,54]. Repeating the above procedure for different harmonic orders, we can then construct the hole dynamics in the ionized molecular ion according to time-frequency mapping underlying the HHG process.

## Data availability
All the data that support the findings of this study are available from the corresponding author upon reasonable request.

## Code availability
All the codes that support the findings of this study are available from the corresponding author upon reasonable request.

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

## Acknowledgements

This work was supported by fundings from the National Key Research and Development Program of China (No. 2019YFA0308300 (L.X.H.), 2017YFE0116600 (P.F.L.)), National Natural Science Foundation of China (No. 91950202 (P.F.L.), 12074136 (L.X.H.), 12021004 (P.X.L.), 11627809 (P.F.L. and P.X.L.), 11934006 (P.F.L. and P.X.L.)), and Natural Science Foundation of Hubei Province (No. (2021CFB330 (L.X.H)). C.D.L. was supported by Chemical Sciences, Geosciences and Biosciences Division, Office of Basic Energy Sciences, Office of Science, U.S. Department of Energy under Grant No. DE-FG02-86ER13491. C.D.L. would like to thank Professor H. Wörner for initial communications on charge migration.

## Author contributions

P.F.L and P.X.L. conceived this research. L.X.H., Y.Q.H., and P.W. performed the experiments. S.Q.S. developed the reconstruction algorithm. L.X.H. and Y.Q.H. performed the simulations. C.D.L., L.X.H., and P.F.L. analyzed the data and wrote the manuscript. B.C.W., X.S.Z., L.L., and W.C. participated in the discussions.

## Competing interests

The authors declare no competing interests.
