## [Peer Review File · Nature Communications]

Filming movies of attosecond charge migration in single molecules with high harmonic spectroscopyReviewers' comments:

Reviewer #1 (Remarks to the Author):

The results presented in the manuscript "Filming movies of attosecond charge migration with high harmonic spectroscopy" belong to attoscience, an important area of modern physics and chemistry. The paper contains both experimental and theoretical results. The presented experimental method (high-harmonic spectroscopy) and theory (high-order harmonic generation) are known, but the novelty of the present work is the method of extraction of information from the data. Experimental state-of-the-art methods combined with molecular alignment and two-color spectroscopy are used. Machine learning algorithm enables retrieval of the complex amplitude and phase of harmonics from single fixed-in-space molecule and construction of a movie of electron migration after tunnel ionization of the molecules at time steps of 50 attoseconds. The angular dependence of molecular charge migration is fully resolved. The presented reconstruction scheme, based on machine learning algorithm and two-color high-harmonic spectroscopy, is a general method for studying ultrafast charge migration in molecules, which paves a way for further advance in tracing and controlling photochemical reactions by femtosecond lasers.

This work is significant for the development of attochemistry. As far as I know, it is original and the dynamics of charge migration on the attosecond time scale is presented for the first time in a movie. I do not see any flaws in the presented results. Therefore, I recommend publication of this manuscript in the present form.

Reviewer #2 (Remarks to the Author):

The authors report on probing attosecond charge migration dynamics in two molecules, N₂ and CO₂, using high harmonic generation (HHG) spectroscopy. Comparing to previous work using HHS spectroscopy, in the current work a machine learning methodology was used to retrieve the molecular dipole moment responsible for the HHG as a function of the angle between the driving field polarization and the molecular axis. Dual color laser fields were used to reduce the timestamp step of the dynamics movie. The authors report "swirl" motions of electrons around the atomic centers which deviates from the "laser direction."

The topic of the work is of intense interest in recent years. The analysis is sound and the conclusion is supported by the results. There are a few issues and some minor problems, as listed below, that won't qualify the current manuscript for publication in Nature Communications.

(1) It is not clear what the authors define as "charge migration," though an electronic wavepacket was created and probed in time. First, dynamics observed is under the influence of external laser field; second, "charge migration" referred in the work by Cederbaum et.al., has a characteristic revival which distinguishes charge migration from charge transfer due to nuclear motion. In this work, Fig 4 (d) shows certain degree of charge return. But overall, it is not clear what was probed is migration or just transfer in the time period of the very first femtosecond.

(2) The authors concluded the advantage of HHS spectroscopy for probing charge migration, which is not specific to their own work. It is not convincing that the method presented in this work has a broader application and can significantly impact the field of studying charge dynamics, since the "complexity of electron wavepacket of cations" will limit this approach to only simple molecular systems. HHS spectroscopy heavily relies on calculations of the HHG source system. The challenge of this approach will remain.

(3) The authors mentioned three challenges in reconstructing electron dynamics and claim that these three issues being resolved in the current work: (1) the reconstruction is an inverse scattering problem; (2) molecules cannot be perfectly aligned; (3) population coefficients of different cation states should be obtained to "extract charge migration" dynamics. These are the nature of the reconstruction of electron wavepacket using HHS and none of them can be changed by the analysis method that is presented in this work.

(4) It is not shown qualitatively or quantitatively how the improvement in the analysis method leads to improved accuracy in "visualizing" charge migration dynamics.

(5) The first pulse is not a pump pulse, because what is probed is not the rotational wavepacket that the first pulse creates. Therefore, the dual-pulse (triple-pulse) experiment is not a pump-probe experiment.

(6) The author should make their description rigorous and accurate. For examples, in the abstract, "along the laser direction" (polarization or propagation?); page 2, work ref [18] published six years ago won't usually be considered "very recent;" page 8, "relative phase between the X and A states." It is the wave functions that have phases.

(7) In page 9, Fig 3(c) appears before Fig 3(a) and (b).

Reviewer #3 (Remarks to the Author):

The reviewed manuscript claims reconstruction of the attosecond hole migration from high-harmonic spectra of two small molecules - carbon dioxide and nitrogen. The technique is claimed to be extensible to larger molecules.

At the core of the work is an algorithm for reconstructing the full (ie magnitude + phase) high-harmonic radiating dipole in the molecular frame from the time evolution of laboratory-frame HHG emission of a coherent rotation wavepacket. This is an elegant and very useful procedure, which seems to be inspired by the work of Varun Makhija on a closely-related problem of reconstructing molecular-frame photoionization amplitudes [e.g. J. Phys. B 54, 145601 (2021)], which I think should be cited.

This part of the work clearly merits publication in some form, subject to a few technical comments listed below.

The reconstructed molecular-frame dipoles are then separated into the "channel" contributions, by multiplying theoretically-calculated channel emission by an orientation-depended "channel amplitude" matrix. A very similar procedure is standard in the field - see e.g. the tutorial by Smirnova and Ivanov at arXiv:1304.2413 (also published as a book chapter), and works cited therein.

Finally, the channel amplitudes are used to calculate the "hole dynamics" in the cations of the target molecules, by weighting the molecular orbitals by the channel amplitudes times the theoretical ionization rates. The resulting "movies" of the hole dynamics constitute the main result of the work. It would however appear that this final step is based on a simple, but fundamental misunderstanding of the nature of hole dynamics.

Specifically, due to the entanglement between the photoelectron and the hole, hole dynamics in the cation exists if, and only if, the continuum part of the wavefunction maintains non-zero overlap between the ion channels [see e.g. PNAS 106, 16556 (2009) for a discussion in the context of high-harmonics spectroscopy]. The issue is very easy to understand: Let us consider the simplest relevant example of a 2-electron, 2-channel system. The total wavefunction at any moment of time is then given by (TeX notation):

$$\Psi(r_1, r_2) = \hat{A} [\Phi_1(r_1) \chi_1(r_2) + \Phi_2(r_2) \chi_2(r_2)]$$

where \hat{A} is the anti-symmetrizer, Φ_i are the orthonormalized "ion" wavefunctions, and χ_i are the wavefunctions of the continuum (or excited electron). For simplicity, we can assume that functions Φ and χ are orthogonal to each other (ie the close-coupling form). Under these assumptions, one can easily calculate the diagonal of the 1-particle density matrix ρ , giving:

$$\rho(r_1) = \sum_i |\Phi_i(r_1)|^2 \langle \chi_i | \chi_i \rangle + \sum_i |\chi_i(r_1)|^2 + 2 \operatorname{Re}[\Phi_1(r_1)\Phi_2(r_1)^* \langle \chi_2 | \chi_1 \rangle]$$

The non-trivial hole dynamics is contained in the last term - which vanishes together with the overlap of the channel continua. For example, for the nitrogen molecule and the laser field oriented along the molecular axis (the case given in Fig. 4f of the main text) the two continua belong to different symmetry irreps, so that their overlap vanishes identically - and no hole dynamics can exist, even in principle. Although the consequences are less drastic for less-symmetric field polarization, they can not be neglected.

Because all the "hole dynamics" discussed in the manuscript is based on the (incorrect) implicit assumption:

$$|\langle \chi_2 | \chi_1 \rangle|^2 = \langle \chi_1 | \chi_1 \rangle \langle \chi_2 | \chi_2 \rangle,$$

the hole dynamics and the fluxes given in Figs. 3 and 4 and discussed in the accompanying text appear to be meaningless. The hole dynamics "movies" referred to in the title also appear to be affected.

I further have a number of more technical comments, in no particular order.

1. The manuscript repeatedly claims that the effects of the alignment distribution on the HHG emission magnitudes and phases have "not been properly treated in all of existing publications". This assertion is manifestly incorrect - coherent averaging over alignment distribution is an essential, and routine, technical step of any HHS treatment of molecules. For a demonstration of its effects, see e.g. the supporting information of Ref. [30] of the manuscript.

2. Captions of Fig. 1 claims a "one-to-one" mapping between time and harmonic photon energy. In the semi-classical treatment including only the short trajectories, this is true - except that the mapping is specific to an emission channel (ie it is NOT the same for the HOMO and HOMO-1 channels, it is also different for channels involving a hole transition). Please correct.

3. Channel-resolved HHS spectroscopy in both carbon dioxide and nitrogen has been extensively studied before, including by some of the articles which are cited by the manuscript in other contexts. Please make sure to clearly cite prior art at the appropriate locations.

4. The results shown in Fig. 3 appear to be wrong. For the electric field polarized along the molecular axis of the CO₂ molecule or CO₂(1+) molecular ion, the total Hamiltonian and the initial conditions conform to the C_{∞v} point group. As the result, the total wavefunction must remain in the same irreducible representation of the group as the initial state. For CO₂, this is Σ^+ , indicating that the total wavefunction, and therefore the total 1-particle density $\langle i \rangle_{\text{must}}$ remain within the Σ^+ irrep, and are fully-symmetric with respect to the rotations around the molecular axis. This symmetry is clearly violated by the panels (a), (b), and (d) - which are therefore unphysical. An exactly the same observation applies to parts of Fig 4. The accompanying discussion (pages 9-10) also appears to be incorrect.

5. The manuscript claims that adding the second-harmonic field at the 0.2% intensity "barely alters the electron dynamics". This may be so - but it is well known to strongly affect the intensities and the phases of high-harmonic emission of even harmonics [See e.g. Nature Physics 2, 781 (2006)]. Yet, from the text and the SI, it would appear that the second-harmonic field is simply disregarded in the treatment of the even harmonics. I believe a clarification is necessary.

6. The reconstruction technique used for $D(\omega, \theta)$ (Eq. 2) appears to completely ignore spatial phase-matching. Yet, it is known to be essential for HHS, and is known to affect different harmonic orders unequally [See e.g. PRL 74, 3776 (1995); PRA 88, 043848 (2013)]. What is the justification for not considering the spatial phase-matching? What is the magnitude of the error this approximation introduces in D ?

7. The phase of the harmonic emission is an experimental observable. Was there any attempt made to measure these phases and to compare them to the (appropriately averaged) reconstructed phases? If not, why not? A measurement of this type would go a long way towards convincing a reader of the correctness and robustness of the reconstruction.

8. It would appear that the continuum-propagation part of the channel dipole \bar{D}_{ij} [Eq. 13] is assumed to depend on the emission energy, but not on the channel. This is known to be generally false [see the tutorial at arXiv:1304.2413, cited above]. What is the justification for this approximation? What is the magnitude of an error it introduces in the coefficients C_{ij} [Eq. 12]?

9. Equations 14 and 15 appear to be doubly-counting the channel ionization yields, because the coefficients C_{ij} are fitted to the experiment, and therefore already include the per-channel ionization probabilities. Furthermore, because coefficients C_{ij} are generally complex [see Eq. 12; both D and $\{\bar{D}\}_{ij}$ are complex], so is the quantity defined by Eq. 15. Yet, orbital population (the diagonal of the 1-particle density matrix in the MO basis) must be a real quantity.

10. Something must have gone wrong with the TDDFT calculations described in the Supplementary figure 11. For symmetry reasons, transitions between $A^2\Pi_g$ and $B^2\Sigma_u^+$ states of $CO_2(+)$ are completely forbidden for the electric field polarized exactly parallel and exactly perpendicular to the molecular axis, for all orders of field interaction and all mechanisms. Yet, the 0-degree (black) curve shows a large transition probability.

11. The technical details given in the Supplementary Note 5 are insufficient to either check or reproduce any of the results there. At the minimum, the laser pulse and the precise definition used for the ion-state amplitudes (formally ill-defined in DFT, due to the lack of the N -electron wavefunction!) must be given.

To summarize, while the manuscript does contain a very interesting and useful kernel (the molecular-frame reconstruction technique for the HHG emission amplitudes), the interpretation of the results is based on a flawed assumption. There also seem to be a large number of technical defects in the work, which manifest in obvious violation of symmetry-mandated conservation laws.

This manuscript is not suitable for publication.

Detailed Replies to the reports by the reviewers

Dear Reviewers,

We have numbered each paragraph of reports from Reviewers 2 and 3 with headings Q1, Q2, ... for easier reference. We also separated their questions into C1, C2 and C3 categories according to

C1— comments and suggested improvement that we both agree

C2—questions we believed are resolved after our clarification

C3— comments made by reviewers that are **not correct**

We comment that Reviewer 3 has raised a number of questions on approaches and results that we have published since 2009. The QRS has been used in all of these original theoretical and experimental papers. Reviewer 3 has referred quite a number of published papers where the authors appear to be from the same group or collaborators. Our two approaches are slightly different and clearly the reviewer is not familiar with our work. We have not made much reference to these earlier publications since this manuscript deals with the retrieval of single-molecule properties from the experimental HHG spectra that has not been treated anywhere in the literature. This is equivalent to solving an inverse scattering problem. To solve such a problem, a simple theory like QRS is needed. We have shown in the Supplementary Materials that from the retrieved molecular data, when convoluted with angular distributions, we can obtain results that are in good agreement with the

measured spectra taken for this work (see Supplementary Fig. 6). Our main point is that theory and experiment used in this manuscript are in good foundation, but these earlier works are not widely cited in this manuscript. For the reviewers, we list our typical publications that are related to this work from the authors:

Selected publications by co-authors of this manuscripts on HHG of CO₂

A. Publications by the theory group at Kansas State

PRL 100, 013903 (08) – first pub. On QRS theory

PRA 80, 013401 (09) -- full story on QRS

PRA 83, 053409 (11) -including propagation

JPB 43, 122001 (10) – **review** article

JPB 49, 053001 (16) -- **Tutorial** on HHG

Graduate text **book** – “Attosecond and strong field Physics...”, (2018)
Cambridge U Press

PRA 101, 013429 (20) - recent – with MIR lasers

PRA 102, 013108 (20) – recent – shaping attosecond pulses with HHG from CO₂

PRL 110, 033006 (13) – HHG experiment and theory for polyatomic molecules

B. Prior experiment and theory collaboration on HHG of CO₂

PRL 121, 163201 (18) – HHG of spinning molecules

PRA 101, 063417 (20)—recent – retrieval of amplitude and phase of HHG

C. Prior publications by the experimental group on HHG of CO₂

Optics Express 28, 21182 (20) – recent – measuring molecular rotational temperature

Optics Express 27, 30172 (19) – measuring molecular echoes by HHG

PRA 99, 053419 (19) – imaging molecular rotation with HHS

PRA 95, 033420 (17) – diffractive molecular-orbital tomography

PRA 91, 043418 (15) – molecular HHG in circularly polarized laser

Optics Express 22, 06362 (14) – probing molecular rotational dynamics with HHG

PRA 89, 013410 (14) – alignment effect on the structural minimum in HHG

In the following, we provided point-to-point response to comments from the three reviewers. Each question asked by the reviewers is followed by our replies. If the question has resulted in a change in the manuscript, how the paper is revised is then provided.

We thank the reviewers for going through these reports.

Reviewer #1 (Remarks to the Author):

The results presented in the manuscript “Filming movies of attosecond charge migration with high harmonic spectroscopy” belong to attoscience, an important area of modern physics and chemistry. The paper contains both experimental and theoretical results. The presented experimental method (high-harmonic spectroscopy) and theory (high-order harmonic generation) are known, but the novelty of the present work is the method of extraction of information from the data. Experimental state-of-the-art methods combined with molecular alignment and two-color spectroscopy are used. Machine learning algorithm enables retrieval of the complex amplitude and phase of harmonics from single fixed-in-space molecule and construction of a movie of electron migration after tunnel ionization of the molecules at time steps of 50 attoseconds. The angular dependence of molecular charge migration is fully resolved. The presented reconstruction scheme, based on machine learning algorithm and two-color high-harmonic spectroscopy, is a general method for studying ultrafast charge migration in molecules, which paves a way for further advance in tracing and controlling photochemical reactions by femtosecond lasers.

This work is significant for the development of attochemistry. As far as I know, it is original and the dynamics of charge migration on the attosecond time scale is presented for the first time in a movie. I do not see any flaws in the presented results. Therefore, I recommend publication of this manuscript in the present form.

Reply: We thank the reviewer for succulently summarizing our

manuscript and for appreciating the important advancement of our work on ultrafast electron dynamics. In this work the experiment has been carried out thoughtfully and the data have been analyzed rigorously. We believe this work deserves to be published in Nature Communications. In the meanwhile, in the following pages, we want to clarify and answer questions raised by the two other reviewers in their previous reports.

Reviewer #2 (Remarks to the Author):

We separate the questions addressed into C1, C2 and C3 according to

C1— comments and suggested improvement that we both agree

C2—questions we believed are resolved after our clarification

C3— comments made by reviewer that are incorrect

For Reviewer 2, we have

C1: Q1, Q7, Q8

C2: Q2, Q3, Q4, Q5, Q6

C3: none

Q1-The authors report on probing attosecond charge migration dynamics in two molecules, N₂ and CO₂, using high harmonic generation (HHG) spectroscopy. Comparing to previous work using HHS spectroscopy, in the current work a machine learning methodology was used to retrieve the molecular dipole moment responsible for the HHG as a function of the angle between the driving field polarization and the molecular axis. Dual color laser fields were used to reduce the timestamp step of the dynamic movie. The authors report “swirl” motions of electrons around the atomic centers which deviates from the “laser direction.”

The topic of the work is of intense interest in recent years. The analysis is sound and the conclusion is supported by the results. There are a few issues and some minor problems, as listed below, that won't qualify the current manuscript for publication in Nature Communications.

Reply: We thank the reviewer for reading our manuscript and that you agreed with us that this work is of intense interest and scientifically sound. We hope that after reading our replies to your questions you will agree with us that our work is suitable (after the revision made) for publication in Nature Communications.

Q2-(1) It is not clear what the authors define as “charge migration,” though an electronic wavepacket was created and probed in time. First, dynamics observed is under the influence of external laser field; second, “charge migration” referred in the work by Cederbaum et.al., has a characteristic revival which distinguishes charge migration from charge transfer due to nuclear motion. In this work, Fig 4 (d) shows certain degree of charge return. But overall, it is not clear what was probed is migration or just transfer in the time period of the very first femtosecond.

Reply: The reviewer is correct in raising this question. Rigorously, the electron dynamics extracted in this work is actually described by a very complicated complex many-electron wave packet. This wave packet can be calculated, but to present it visually as a movie, the modulus square of the wave packet is presented and its change with time is called charge migration previously in the literature. In this work, we demonstrated how to extract charge migration from experimental HHG spectra. Since many-electron density has very high dimensions, the ion core of a molecule like N_2 and CO_2 is viewed as a hole such that a movie can be presented. Hole migration was first introduced theoretically in L. S.

Cederbaum and J. Zobeley, Chem. Phys. Lett. 307, 205 (1999) without addressing on how they can be “measured”. The so-called HHG spectroscopy has been proposed as a means to extract short-time electron dynamics as described in our manuscript and the first experiment was reported in Kraus et al, Science 350, 790 (2015) on HCCI molecules for molecules oriented parallel and perpendicular to laser polarization. As discussed in our manuscript, in that work they took the HHG spectra measured along each orientation direction as the HHG generated by each single fixed-in-space molecule along that direction. This procedure is clearly incorrect when molecules are only partially aligned, but the approximation has been used in every HHG spectroscopy paper because there is no simple way to do otherwise. In our work, for the first time we have demonstrated how this can be done and what measurements should be made. This manuscript reports what we have accomplished.

Since electron dynamics at early times is presented in terms of charge migration--the modulus square of the electron wave packet, and charge migration has been widely used in the literature, we prefer not to adopt a new nomenclature. This work is to show how to obtain many-electron wave packet, and in the present case, a single hole wave packet, such that they can be viewed as a “movie”. The movie in our manuscript does show viewers how the single hole evolves in space in the first few tens to hundreds of attoseconds after one electron has been removed from the molecule. We followed the literature and Kraus et al [Science 350, 790 (2015)] and call the movie as charge migration. No nuclear motion is expected in such a short time, nor that the hole is expected to stop at the

end of the movie shown. What one can see is how the hole “migrates” from one side to another, or more accurately in the case of CO₂, to swirl from one side to the other and back, and is not necessarily along the molecular axis or the polarization axis. The swing or swirl motion in the case of CO₂ occurs in tens of attoseconds.

Q3-(2) The authors concluded the advantage of HHS spectroscopy for probing charge migration, which is not specific to their own work. It is not convincing that the method presented in this work has a broader application and can significantly impact the field of studying charge dynamics, since the “complexity of electron wavepacket of cations” will limit this approach to only simple molecular systems. HHS spectroscopy heavily relies on calculations of the HHG source system. The challenge of this approach will remain.

Reply: We have not said that HHG spectroscopy is specific to our work. It has been “talked” about before, but we can claim that we are the first to do HHG spectroscopy correctly such that charge migration in a molecule can be extracted from the experimental HHG spectra.

We agree that it is not realistic to expect the method be extended to more complex molecules since HHG experiments are few for complex molecules and that a complex molecule has tons of parameters to be specified. On the other hand, there are no other ways beyond HHG spectroscopy to probe electron dynamics at their natural attoseconds timescale. A demonstration of attosecond charge migration from the experiment is surely of interest even if it is only for small molecules.

In our reconstruction, HHG calculations are based on the QRS theory. The QRS theory has been widely applied in the field of molecular HHG beyond simple molecules like N₂, CO₂, O₂, but can be applied to many other slightly more complex molecules. However, as explained in our manuscript, the implementation of HHG spectroscopy, as well as any other methods, may be limited for a large molecule because of the complexity of the electron wave packet.

Looking ahead, we think the method used here can be extended to a few slightly more complex molecules, for example, some small hydrocarbon molecules, to probe how the ring structure affects the attosecond electron dynamics. Such experiments are being evaluated and hopefully can be carried out in the coming months.

Q4-(3) The authors mentioned three challenges in reconstructing electron dynamics and claim that these three issues being resolved in the current work: (1) the reconstruction is an inverse scattering problem; (2) molecules cannot be perfectly aligned; (3) population coefficients of different cation states should be obtained to “extract charge migration” dynamics. These are the nature of the reconstruction of electron wavepacket using HHS and none of them can be changed by the analysis method that is presented in this work.

Reply: We have not expected this comment. Perhaps this is because of the wording in the last paragraph of the Introduction (“*Here we demonstrate that, by resolving the above three issues...*”). The three issues are inherent to the problem and cannot be “resolved”. With proper

theoretical and computational tools, as presented in the present work for simple molecules like N₂ and CO₂, the reconstruction can be carried out. We have changed the mentioned sentence to avoid the confusion.

Revision: The sentence “*Here we demonstrate that, by resolving the above three issues...*” in the last paragraph of the Introduction has been changed to “*Here we demonstrate that, by properly handling the above three issues...*”. (page 3 of the main text)

Q5-(4) It is not shown qualitatively or quantitatively how the improvement in the analysis method leads to improved accuracy in “visualizing” charge migration dynamics.

Reply: We had stated qualitatively in the main text (see Lines 8-11 in the 3rd paragraph of the Introduction) and demonstrated quantitatively in Supplementary Fig. 3 that the tools we introduced in this paper is needed for the accurate retrieval of charge migration from HHG spectra. The most important improvement is that we can extract single-molecule harmonic dipoles correctly from the experimental data. In almost all previous HHS experiments, a long prevailing incorrect practice is to take single-molecule harmonic amplitude and phase directly from the experimental data which are the results of averaging over many molecules. In Supplementary Fig. 3, we showed how wrong this practice is, especially for the phase of a single molecule dipole. It is well known that phases determine the dynamics in the time domain. Using angle-averaged experimental dipole to represent the single-molecule dipole would introduce large errors in charge migration. This is the

method used in Kraus et al. Thus, without the processing of experimental data as presented in our work, the extracted results would be incorrect (not only imprecise).

Q6-(5) The first pulse is not a pump pulse, because what is probed is not the rotational wavepacket that the first pulse creates. Therefore, the dual-pulse (triple-pulse) experiment is not a pump-probe experiment.

Reply: The term “pump-probe” in our original manuscript may be less felicitous. We have rephrased it in the revised manuscript.

Q7-(6) The author should make their description rigorous and accurate. For examples, in the abstract, “along the laser direction” (polarization or propagation?); page 2, work ref [18] published six years ago won’t usually be considered “very recent;” page 8, “relative phase between the X and A states.” It is the wave functions that have phases.

Reply: The descriptions mentioned by the reviewer have been corrected in the revised manuscript. We have also checked throughout the whole manuscript to ensure our descriptions are accurate.

Q8-(7) In page 9, Fig 3(c) appears before Fig 3(a) and (b).

Reply: We have revised the manuscript such that all the figures are cited in order.

Reviewer #3 (Remarks to the Author):

We separate the questions addressed into C1, C2 and C3 according to

C1— comments and suggested improvement that we both agree

C2—questions we believed are resolved after our clarification

C3— comments made by reviewer that are incorrect

For Reviewer 3, we have

C1: none

C2: Q4, Q5, Q7, Q8, Q9, Q10, Q12, Q13

C3: Q1, Q2, Q3, Q6, Q11, Q14

Authors' Remarks to Reviewer 3: We urge the reviewer to pay special attention to questions listed in C3 where we have strong disputes with your criticisms. Key phrases of your main criticism are highlighted in yellow color.

The reviewed manuscript claims reconstruction of the attosecond hole migration from high-harmonic spectra of two small molecules - carbon dioxide and nitrogen. The technique is claimed to be extensible to larger molecules.

Q1-At the core of the work is an algorithm for reconstructing the full (i.e. magnitude + phase) high-harmonic radiating dipole in the molecular frame from the time evolution of laboratory-frame HHG emission of a coherent rotation wavepacket. This is an elegant and very useful procedure, which seems to be inspired by the work of Varun Makhija on a closely-related problem of reconstructing molecular-frame

photoionization amplitudes [e.g. J. Phys. B 54, 145601 (2021)], which I think should be cited.

Reply: It is unthinkable that the reviewer began the report by saying that our work was inspired by the cited paper of Varun Makhija. By alluding that we copy the idea of Makhija in our work, the reviewer has injected a very serious negative impression to the editor from the very beginning. It is truly unfair to the authors since we have put a great effort into this work and it was not “inspired” by the mentioned paper. It can be easily checked.

First, Makhija paper appeared on 16 August, 2021 while the reviewer probably received our manuscript in late August or early September since we submitted it to Nature Communications on 11 August, 2021. [In fact, our manuscript was first submitted to Nature sequence on 29 January, posted at Research Square (<https://www.researchsquare.com/article/rs-185507/v1>) on 02 March, 2021]. What motivated the reviewer to make such an erroneous statement and what is the purpose?

Second, if the reviewer has read our paper and Makhija’s paper, he/she should have realized the two papers are dealing with entirely different issues. In Makhija’s paper, they want to obtain photoelectron angular distributions in the molecular frame using angular distributions from partially aligned molecules in the laboratory frame. In this case, the quantities involved are all positive-definite. In our work, we are to retrieve complex-valued single-molecule laser-induced dipole in the molecular frame using harmonic signal generated from partially aligned molecules. In our case, we need to extract the amplitude and phase of the

dipole at each angle. Retrieving the phase is much more challenging and the two methods used are completely different.

In reviewing a manuscript, the reviewer should make statements based on facts, especially the negative ones.

This part of the work clearly merits publication in some form, subject to a few technical comments listed below.

Q2-The reconstructed molecular-frame dipoles are then separated into the "channel" contributions, by multiplying theoretically-calculated channel emission by an orientation-depended "channel amplitude" matrix. A very similar procedure is standard in the field - see e.g. the tutorial by Smirnova and Ivanov at arXiv:1304.2413 (also published as a book chapter), and works cited therein.

Finally, the channel amplitudes are used to calculate the "hole dynamics" in the cations of the target molecules, by weighting the molecular orbitals by the channel amplitudes times the theoretical ionization rates. The resulting "movies" of the hole dynamics constitute the main result of the work. It would however appear that **this final step is based on a simple, but fundamental misunderstanding of the nature of hole dynamics.**

Specifically, due to the entanglement between the photoelectron and the hole, hole dynamics in the cation exists if, and only if, the continuum part of the wavefunction maintains non-zero overlap between the ion channels [see e.g. PNAS 106, 16556 (2009) for a discussion in the context of high-harmonics spectroscopy]. The issue is very easy to understand: Let

us consider the simplest relevant example of a 2-electron, 2-channel system. The total wavefunction at any moment of time is then given by (TeX notation):

$$\Psi(r_1, r_2) = \hat{A} [\Phi_1(r_1) \chi_1(r_1) + \Phi_2(r_2) \chi_2(r_2)] \quad (1)$$

where \hat{A} is the anti-symmetrizer, Φ_i are the orthonormalized "ion" wavefunctions, and χ_i are the wavefunctions of the continuum (or excited electron). For simplicity, we can assume that functions Φ and χ are orthogonal to each other (ie the close-coupling form). Under these assumptions, one can easily calculate the diagonal of the 1-particle density matrix ρ , giving:

$$\begin{aligned} \rho(r_1) &= \sum_i |\Phi_i(r_1)|^2 \langle \chi_i | \chi_i \rangle + \sum_i |\chi_i(r_1)|^2 \\ &+ 2 \operatorname{Re} [\Phi_1(r_1) \Phi_2^*(r_1) \langle \chi_2 | \chi_1 \rangle] \end{aligned} \quad (2)$$

The non-trivial hole dynamics is contained in the last term - which vanishes together with the overlap of the channel continua. For example, for the nitrogen molecule and the laser field oriented along the molecular axis (the case given in Fig. 4f of the main text) the two continua belong to different symmetry irreps, so that their overlap vanishes identically - and `_no_hole` dynamics can exist, even in principle. Although the consequences are less drastic for less-symmetric field polarization, they can not be neglected.

Because all the "hole dynamics" discussed in the manuscript is based on

the (incorrect) implicit assumption:

$$|\langle \chi_2 | \chi_1 \rangle|^2 = \langle \chi_1 | \chi_1 \rangle \langle \chi_2 | \chi_2 \rangle,$$

$$|\langle \chi_1 | \chi_2 \rangle|^2 = \langle \chi_1 | \chi_1 \rangle \langle \chi_2 | \chi_2 \rangle \quad (3)$$

the hole dynamics and the fluxes given in Figs. 3 and 4 and discussed in the accompanying text appear to be meaningless. The hole dynamics "movies" referred to in the title also appear to be affected.

Reply:

[1] To begin with, in the first paragraph, it should be noted that Smirnova and Ivanov at arXiv:1304.2413 (also published as a book chapter) and other references have never been able to reconstruct the molecular frame dipole.

[2] In the next paragraph, the reviewer said "*this final step is based on a simple, but fundamental misunderstanding of the nature of hole dynamics*" and then gave a simple example using three equations, with a "simple" example for a two-electron system. The issue has to do with the entanglement of particles in a many-body system. However, there are a number of issues in what the reviewer said that follows:

(a) Eq. (1) is obviously incorrect. You cannot use r_1 for both the ion and the continuum electron.

(b) The bracket $\langle | \rangle$ between two continuum functions χ is not defined since it would diverge.

(c) Why 1-particle density from an entangled system has anything to do with ion (or hole) density which is defined only when the continuum electron is decoupled from the ion.

In addition, the example in Eq. (1) is meaningless since you cannot write down the wave packet in an ionization event of a two- or more particle systems like Eq. (1), and one cannot talk about the wave packet of a subsystem alone if the subsystem is still coupled with the other subsystem. Thus, the example given by the reviewer is meaningless and wrong. Even though the reviewer has cited PNAS 106, 16556 (2009) to make his/her point, in that paper the hole dynamics is also referred to the condition that the continuum electron is decoupled from the ion, which is the same model used by others in HHG spectroscopy and by Kraus et al. (Science 350, 790 (2015)). Without such an approximation, it is not possible to talk about ion (or hole) dynamics, and charge migration.

[3] The symmetry argument by the reviewer is also wrong since symmetry does not apply to the subsystem. (In quantum mechanics, rigorously speaking, each individual particle has no physical meaning, thus symmetry says nothing about the subsystem.) In HHG, an electron can be removed from different orbitals by the laser such that holes of different symmetry are created and symmetry is no longer applied to the ion (hole).

We have checked Smirnova and Ivanov at arXiv:1304.2413 and PNAS 106, 16556 (2009), the derivation uses so many approximations, making it intractable. Clearly in dealing with a many-electron system, any statement about individual orbitals is an approximation. Our approach in QRS and in Eqs. (S1-S6) explicitly show how we obtain hole dynamics. Simply speaking, in doing so, we have to decouple the core from the electron that has been released from the molecule, and since more than

one core levels are affected, a wave packet is created. This wave packet is modified in the laser field until the “free” electron returns to recombine with the ion and emits the harmonic. This model, together with the QRS theory, allows us to calculate harmonic spectra. Such calculations have been shown in good agreement with experiments in many publications. We do not need to use the complicated models cited by the reviewer. Since this paper is not about calculating harmonic generation itself (which we have studied extensively about 10 years ago), but rather to extract the amplitude and phase of the hole state as a function of time in the laser field, our retrieval method relies on the QRS theory. The wave packet consists of a few holes, is represented by the complex amplitudes. Since complex amplitudes are abstract, the modulus square of the electron wave packet, or charge density, as a function of time, is used to mean “charge migration”. This is how charge migration was defined by L. S. Cederbaum and J. Zobeley, *Chem. Phys. Lett.* 307, 205 (1999), by Kraus et al. (*Science* 350, 790 (2015) and also in a review at *Angew. Chem. Int. Ed.* 57, 5228-5247 (2018). The only way to talk about hole dynamics is to use the same approximation, treating the released electron and the ion being not entangled. This model is valid when the “continuum” electron is not very close to the ion.

In summary, the reviewer misunderstood completely the theory model presented in our work. The reviewer’s questions and comments are based on a fundamental assumption that the electronic dynamics in the cation are quasi static. This assumption has been used in previous works (including the tutorial by Smirnova and Ivanov at arXiv:1304.2413 and

PNAS 106, 16556 (2009), and the related references therein) on HHG from CO₂. However, this assumption is evidently untenable in the HHG process, since an intense external laser field is present. In the strong laser field, ionization of the molecule to the ground or excited states of the cation will prepare one-hole configurations in the HOMO and HOMO-1 etc of the neutral molecule (i.e., Eq. 9 in our manuscript). In other words, the “ion” wavefunction is a linear combination of eigenstates of the ion, the time-dependent evolution of the ion wavefunction thus evolves in time without constraints of the symmetry. To sum up, in our work, we have reconstructed the hole dynamics in the external laser field. It is not the field-free case. The theoretical description of the hole dynamics in our work is same to that in the prior work by Kraus et al. (Science 350, 790 (2015) and also a review at Angew. Chem. Int. Ed. 57, 5228-5247 (2018)). There is nothing wrong in our results.

In summary, we refute the negative comments by the reviewer and stand up to what we have done.

I further have a number of more technical comments, in no particular order.

Q3-1. The manuscript repeatedly claims that the effects of the alignment distribution on the HHG emission magnitudes and phases have “not been properly treated in all of existing publications”. **This assertion is manifestly incorrect** - coherent averaging over alignment distribution is an essential, and routine, technical step of any HHS treatment of molecules. For a demonstration of its effects, see e.g. the supporting

information of Ref. [30] of the manuscript.

Reply: The reviewer has quoted our statement out of context. What we said in our manuscript is that the second step---“*extracting fixed-in-space single-molecule harmonic amplitudes and phases*”---“*has not been properly treated in all of the existing publications*”. This is true, because the coherent averaging over molecular alignment in HHG process makes the inverse reconstruction with the iterative algorithms very difficult and has been avoided in **all** prior published experimental papers. Our work is the first time where fixed-in-space single-molecule dipole moments have been successfully retrieved from experiment. The reference [30] mentioned by the reviewer only has forward calculation where single molecule dipoles are averaged over the angular distribution in order to compare with the experiment. The authors there have never been able to solve this complicated inverse problem.

Q4-2. Captions of Fig. 1 claims a “one-to-one” mapping between time and harmonic photon energy. In the semi-classical treatment including only the short trajectories, this is true - except that the mapping is specific to an emission channel (ie it is NOT the same for the HOMO and HOMO-1 channels, it is also different for channels involving a hole transition). Please correct.

Reply: We have revised the captions of Fig. 1 to make the description more accurate.

Revision: In the captions of Fig. 1, the word “one-to-one” has been removed. [Page 4 of the main text]

Q5-3. Channel-resolved HHS spectroscopy in both carbon dioxide and nitrogen has been extensively studied before, including by some of the articles which are cited by the manuscript in other contexts. Please make sure to clearly cite prior art at the appropriate locations.

Reply: We have carefully checked the references in our manuscript so they are now cited appropriately.

Q6-4. The results shown in Fig. 3 appear to be wrong. For the electric field polarized along the molecular axis of the CO₂ molecule or CO₂⁺ molecular ion, the total Hamiltonian and the initial conditions conform to the C_{∞v} point group. As the result, the total wavefunction must remain in the same irreducible representation of the group as the initial state. For CO₂, this is Σ^+ , indicating that the total wavefunction, and therefore the total 1-particle density *must* remain within the Σ^+ irrep, and are fully-symmetric with respect to the rotations around the molecular axis. This symmetry is clearly violated by the panels (a), (b), and (d) - which are therefore unphysical. An exactly the same observation applies to parts of Fig 4. The accompanying discussion (pages 9-10) also appears to be incorrect.

Reply: As the reviewer said correctly, the symmetry should be preserved by the total wavefunction, but charge migration is for the molecular ion only, i.e., it is the hole dynamics. The initial ionization of the molecule by the laser creates a hole that contains ions of different symmetries. The resulting wave packet further evolves in time in the presence of the laser

field, and the modulus square of the wave packet is called charge migration of the hole dynamics. Alternatively, if only HOMO is ionized, then the symmetry would be preserved. In that case, there would be no hole dynamics. Using physics incorrectly, it is surprising that the reviewer is so confident to claim that our main results in Figs. 3 and 4 are incorrect. Such strong incorrect statements definitely would have negatively influenced the editor's judgement on this work. It is not a good practice to use "appears to be wrong" or "appears to be incorrect" in a reviewer report if one is not sure. Asking the authors to clarify is the more proper way. While a reviewer is regarded as an expert, in general the authors would know better about their work. It is better to ask a question first instead of saying what has been done is incorrect unless the reviewer is 100% sure.

Alternatively, in more precise terms, one notes that strong field ionization will prepare configurationally mixed states. For CO₂, for laser field polarized along the molecular axis, electrons can be ionized with comparable rates from both the HOMO and HOMO-2 orbitals. Since the HOMO and HOMO-2 have different symmetries in space (in fact, HOMO-1 is also involved due to the laser-induced transition), the total ion core wavefunction and thus the hole density cannot be fully-symmetric with respect to the molecular axis. We emphasize again that we have reconstructed the hole dynamics in the external laser field, not the field-free case, and nothing is wrong in our results.

Q7-5. The manuscript claims that adding the second-harmonic field at the

0.2% intensity “barely alters the electron dynamics”. This may be so - but it is well known to strongly affect the intensities and the phases of high-harmonic emission of even harmonics [See e.g. Nature Physics 2, 781 (2006)]. Yet, from the text and the SI, it would appear that the second-harmonic field is simply disregarded in the treatment of the even harmonics. I believe a clarification is necessary.

Reply: What the reviewer has said in the first half of the paragraph is correct. Figure 2a shows exactly that a weak second harmonic can greatly affect the even harmonics which are essential for our reconstruction process. Perhaps our original wording had not emphasized this. We have revised the narrative in the manuscript to state this fact more prominently.

Revision: In the revised manuscript, the sentence “*The SH field is weak ($\sim 2 \times 10^{-3}$ of the fundamental field) such that it barely alters the electron dynamics of the molecular ion (see Supplementary Note 5)*” has been changed to “*The SH field is weak ($\sim 2 \times 10^{-3}$ of the fundamental field), which barely alters the electron dynamics of the molecular ion (see Supplementary Note 5), but could affect the intensities and phases of even harmonics as in [41]*” to emphasize this point (see Lines 6-8 in the 2nd paragraph on Page 7).

Q8-6. The reconstruction technique used for $D(\omega, \theta)$ (Eq. 2) appears to completely ignore spatial phase-matching. Yet, it is known to be essential for HHS, and is known to affect different harmonic orders unequally [See e.g. PRL 74, 3776 (1995); PRA 88, 043848 (2013)]. What is the justification for not considering the spatial phase-matching? What is the

magnitude of the error this approximation introduces in D?

Reply: We do understand the phase matching issues. In our experiment, we used a very small background gas pressure (~20 torr). The macroscopic (spatial) phase-matching effect is rather weak, which should hardly affect our reconstruction result. To demonstrate this point, we have plotted in Fig. R1 the HHG signal of H21 of N₂ measured in the one-color experiment. Figure R1(a) shows the spatial profile of H21. Figure R1(b) plots the time-dependent signals of H21 integrated over different spatial ranges indicated by the respective rectangles in (a). One can see the time-dependent HHG yields are highly consistent for signals integrated over the on-axis (104-120 pixel), the off-axis (128-143 pixel) and the whole regions (72-156 pixel). Similar results have been found for other harmonic orders. This result demonstrates good spatial phase-matching for all harmonics in our experiment. In our work, the whole space integration results were used in the reconstruction.

Figure R1: (a) Spatial profile of H21 of N₂ measured in the one-color experiment. (b) Time-dependent HHG yield of H21 integrated over different spatial ranges indicated by the rectangles in (a). Squares, diamonds and circles correspond to the on-axis (104-120 pixel), off-axis (128-143 pixel) and whole space (72-156 pixel) integrations, respectively. Here the harmonic yields have been normalized to the results measured for the isotropic case.

Q9-7. The phase of the harmonic emission is an experimental observable. Was there any attempt made to measure these phases and to compare them to the (appropriately averaged) reconstructed phases? If not, why not? A measurement of this type would go a long way towards convincing a reader of the correctness and robustness of the reconstruction.

Reply: What the reviewer suggested would require a separate setup which is not available in our laboratory and it is not a practical method to “convince” a reader of the correctness and robustness of the reconstruction. The latter is best tested at the reconstruction steps. The reviewer can find such tests using HHG spectra generated from the QRS theory (tested in many previous publications) that are compared to those obtained from the retrieved data in Supplementary Figs. 4, 5 and 6. While one can do RABBIT measurement to determine the phases, in reality it is not a simple setup. For such experiment, one also needs to generate stronger harmonic signals, for example, by increasing the gas pressure, but in doing so, the phase matching could be modified. With the low pressure used in our experiment, as we explained in Q8, phase matching is easier to achieve. In fact, Supplementary Fig. 6 has shown that the time-dependent HHG spectra measured and those constructed from the retrieved data agree with each other quite well. Since the time-dependent spectra in the experiment involves angular average over the complex-valued single molecule dipoles, to some degree it amounts to confirming that the phase has been accurately retrieved as well. We believe that such tests are actually better since it is rather difficult to

generate harmonics exactly under the same condition if the data are taken from separate measurements.

Q10-8. It would appear that the continuum-propagation part of the channel dipole \bar{D}_{ij} [Eq. 13] is assumed to depend on the emission energy, but not on the channel. This is known to be generally false [see the tutorial at arXiv:1304.2413, cited above]. What is the justification for this approximation? What is the magnitude of an error it introduces in the coefficients C_{ij} [Eq. 12]?

Reply: The reviewer is right. The continuum-propagation part (a_{acc}) of the dipole \bar{D}_{ij} does depend on the channel and it is included in the QRS calculation. It was an oversight in the previous version.

Revision: We have added a superscript i in a_{acc} in Eqs. (7), (11), (13) in the revised manuscript to distinguish different channels.

Q11-9. Equations 14 and 15 appear to be doubly-counting the channel ionization yields, because the coefficients C_{ij} are fitted to the experiment, and therefore already include the per-channel ionization probabilities. Furthermore, because coefficients C_{ij} are generally complex [see Eq. 12; both D and \bar{D}_{ij} are complex], so is the quantity defined by Eq. 15. Yet, orbital population (the diagonal of the 1-particle density matrix in the MO basis) must be a real quantity.

Reply: No, we did not double count. In our calculations, the ionization yield is included in the ionization part of the dipole moment D . Thus, in

Eqs. 14 and 15, the the channel ionization yields are not doubly-counted. To avoid confusion, we have rewritten Eq. 8 in the revised manuscript. On the other hand, the orbital populations in our reconstruction are given by the modulus square of p_j defined in Eq. (15).

Revision: We have added the ionization rate η_i to the right-hand side of Eq. (8) in the revised manuscript.

Q12-10. Something must have gone wrong with the TDDFT calculations described in the Supplementary figure 11. For symmetry reasons, transitions between $A^2\Pi_g^+$ and $B^2\Sigma_u^+$ states of CO_2^+ are completely forbidden for the electric field polarized exactly parallel and exactly perpendicular to the molecular axis, for all orders of field interaction and all mechanisms. Yet, the 0-degree (black) curve shows a large transition probability.

Reply: We have checked our TDDFT procedure and nothing is wrong in the calculations. We did find a mistake in the description of the Supplementary figure 11, which shows the transition between $A^2\Pi_g^+$ and $X^2\Pi_u^+$ states of CO_2^+ , not $A^2\Pi_g^+$ and $B^2\Sigma_u^+$ states. We have corrected it in the revised Supplementary materials. Thanks for bringing it out to our attention.

Revision: We have revised the description of the Supplementary figure 11 (see the last paragraph on Page 20 of the Supplementary materials).

Q13-11. The technical details given in the Supplementary Note 5 are insufficient to either check or reproduce any of the results there. At the

minimum, the laser pulse and the precise definition used for the ion-state amplitudes (formally ill-defined in DFT, due to the lack of the N-electron wavefunction!) must be given.

Reply: The laser pulse and the definition of the ion-state amplitudes used in our TDDFT simulations have been clarified in the revised Supplementary materials.

Revision: The ion-state amplitudes have been defined at the end of the paragraph on Page 17 of the Supplementary materials. The laser field used in our TDDFT calculations has also been given on Page 18 of the Supplementary materials.

Q14-Final Summary by Reviewer 3. To summarize, while the manuscript does contain a very interesting and useful kernel (the molecular-frame reconstruction technique for the HHG emission amplitudes), the interpretation of the results is based on a flawed assumption. There also seem to be a large number of technical defects in the work, which manifest in obvious violation of symmetry-mandated conservation laws. This manuscript is not suitable for publication.

Final Summary of our Reply: First, we want to thank Reviewer 3 for your careful reading of our manuscript and for asking many questions. Some of these questions are helpful that guide us to improve the presentation. However, we are saddened by several of your severe misunderstanding on the important technical issues in our work. We have made our best effort to explain to you where we disagree with you. In the final paragraph you concluded that in our work “*the interpretation of the*

results is based on a flawed assumption. There also seem to be a large number of technical defects in the work, which manifest in obvious violation of symmetry-mandated conservation laws.” From our replies to your questions Q1, Q2, Q3, Q6 and Q11, **the incorrect interpretation is on your side**. The criticism on “serious violation of symmetry-mandated conservation laws” is **due to your misunderstanding of symmetry in a sub-system versus the whole system in a many-body problem**. We stand firm to stress that our work has been built on solid foundation.

From our side, this manuscript is a cumulation of our experimental and theoretical works on HHG for the last 15 years. For the first time, we have been able to disentangle harmonics from single molecules directly from the experimentally measured harmonics. This has never been done in the literature despite that HHG has been studied for more than 30 years. The basic idea of HHG is simple, but to do it right is not easy. Great effort has been made in order to get the results reported in our manuscript.

We do not know if you would concur with our rebuttal to your criticism. If you still disagree with us, we urge you help making our “quarrels” open to the public by making positive recommendation to the editor. Since Nature journals allow reviewer reports to appear as Comment to the published paper, this would be a good forum to present your opinion on this work. That would be fair to authors and fair to the reviewer. The readers would have the chance to find out where the dispute is. Such dispute is healthy for the field.

Finally, just a remark, the reconstruction of single molecule dipole

presented in this work also has significant implications to all time-delay experiments and theories in ultrafast field for molecular targets. Such problems remain to be addressed by researchers in the future.

REVIEWER COMMENTS

Reviewer #1 (Remarks to the Author):

The Editor's question is:

"During the last round of review, one of the other reviewers (Reviewer #3) had some technical concerns about the method used by the authors and this has impact on the validity of the method. We are hoping that you might be willing to assist us further by taking a look at Reviewer #3's comments on the manuscript and the authors response to them. We would be interested in the extent to which Reviewer #3's position is justified and the extent to which the authors have dealt with their comments. You can see these files by using the link provided below."

Answer: The relevant part is Q2 of the 3rd referee, which makes questionable the theoretical method used in the manuscript. Question is related to the entanglement between the photoelectron and the hole in the HHG process. The reference PNAS 106, 16556 (2009) is mentioned, as well as a tutorial by Smirnova and Ivanov at arXiv:1304.2413 (also published as a book chapter). A "simple" example is given to show that the method used in the manuscript is questionable/meaningless. The authors, of course, disagree and explain why they think that the referee's argumentation is wrong.

My opinion is that the issue with which both the authors and the referee deal is not connected to the present manuscript. It is about which theoretical approach to HHG by a molecular system should be used. One is the QRS theory developed by the authors starting in 2008 for atoms and applied latter to molecules. The other one is the electron-hole dynamics of the above-mentioned PNAS and arXiv papers (and many more). Both theories are approximative and it is difficult to follow all approximations. Theories are confirmed by comparison with the experimental results. In fact, both mentioned theories were "confirmed" by experiments. But the HHG experiments are complicated because of the macroscopic effects and this "confirmation" is only approximative.

So, the "fight" whose theory is better should be solved finding a problem where these theories give different results and the experiment will decide that (for the parameters of this experiment) one or the other (or neither of them) is correct.

To conclude, I think that this manuscript should be published. It has both theory and experiment and I do not see any flaws or problems.

Reviewer #2 (Remarks to the Author):

Any publications in professional scientific journals should be written in a rigorous manner, particularly about definition of the phenomenon of interest. The central interest of this article is probing charge migration and a data interpretation method using ML for extracting electron migration dynamic information from HHS data. Electronic wavepacket based on multi-electron correlation probed by HHS specifically regards electron dynamics associated with the recombination process, which is a strong field phenomenon. As the authors also mentioned, the electron dynamics probed by HHS are under the influence of the strong field (laser field). The concept of “charge migration” in Cederbaum’s work is field-free superposition of electronic states due to electron correlation, and therefore is widely cited in statements of its ramifications in chemical reactions and biological systems where electron correlations occur in absence of strong fields. A signature character of the “migration” introduced by Cederbaum, et al., which inspired numerous theoretical and experimental efforts even up till today is the revival of the electron density distribution. Not all electronic wavepackets are due to electron correlations and hence lead to “migration” by its origin. Being bound to strong field and limited in probing time range are well known limitations of HHS technique. Simple experimental setup of HHS is associated with challenge of data interpretation which heavily relies on theoretical work. I don’t mean that any work improving HHS is of little value because of the intrinsic drawbacks of the technique. Authors’ claim that the ML-based HHG interpretation method described in this article establishes a general reconstruction scheme for studying ultrafast charge migration is vague and overstating.

Cederbaum’s work was considered by many (see citations) as original work that opened up the research avenue of investigating electron correlation induced charge dynamics (charge migration). As the authors explained, they used this key term in a loose way because they prefer not to adopt a new nomenclature. No clarification of this “adaption” in the original draft. In the revised draft, the authors took no effort to address this confusion due to the shifted meaning or question which would be likely raised by other readers who are familiar with the history of work regarding charge migration.

There are other terms that the authors seemingly self-defined and used in a “casual” way, for example, using attosecond photoemission for streaking technique, photo-fragmentation spectroscopy for mass/ion spectroscopy.

I cannot agree with the authors’ statement that “there are no other ways beyond HHG spectroscopy to probe electron dynamics at their natural attoseconds timescale.” RABBIT and energy/angular streaking are common experimental techniques for probing sub fs dynamics, each with their own merits and limits. Isolated pulses of several tens of as duration can be generated via HHG now. Energy streaking directly measures electron energy and when combined with reaction microscope, can do angular dependent study in molecular frame without the need of molecular alignment.

The authors' response to comment (4) is off the track. They argued that "We had stated qualitatively in the main text (see Lines 8-11 in the 3rd paragraph of the Introduction) and demonstrated quantitatively in Supplementary Fig. 3 that the tools we introduced in this paper is needed for the accurate retrieval of charge migration from HHG spectra." First, there is no such a section "Introduction" in the manuscript. Second, it is not clear what the authors meant by "stated qualitatively;" words are words. Thirdly, Fig. 3 in the supplement materials depicts dipole moments. What is missing (point in comment 4) is how does it translate into illustrating the "migration" dynamics (evolving spatial distribution). The long argument responding to a referee without making any changes in the manuscript simply doesn't improve the clarity and readability of the article. And the writing of this article in these aspects is far from perfection.

This work has solid results and shows a new method. It is expected that the new method can improve HHS data interpretation which will reveal interesting information of certain processes that cannot be accessed with conventional HHS data analysis. It is not convincing that it will have a strong general impact in the corresponding field, though the topic this work concerns is of intense interest. It is certainly worth publication, but not in Nature Communications, in my opinion. The manuscript needs to be improved as pointed out above for publication.

Reviewer #3 (Remarks to the Author):

First of all, I apologize to the authors, who seem to have taken my original report, which was intended strictly as a opinion on the `_physics_`, as a `_personal_` attack. Nothing could be further from the truth, and I deeply regret that my completely frank and admittedly robust opinion - something which I believe is the point of anonymous refereeing - seems to have caused the authors such a personal offence. Once again, I do sincerely apologize.

Returning to the physics: To my great disappointment, the revised manuscript does not address any of the major issues brought up in my original report. Upon carefully reading the authors' rebuttal, I remain convinced that my original arguments are completely valid. I am most grateful to the authors for pointing out the obvious typing mistakes in some of the formulae I gave: it is of course true that the coordinate index should be changed in one of the functions. Typing TeX "blind", without rendering the formulae is unfortunately error-prone.

While the suggestion by the authors that I should recommend publications of the results I believe to be wrong, alongside the explanation of why, is certainly enticing, I must respectfully decline the offer. It would not be fair to make public comments anonymously - and the authors' rebuttal seems to imply that the authors do intend to make it a personal fight, rather than a discussion of physics.

Upon careful reconsideration, my original assessment and recommendation therefore stand.

Replies to the reports by the reviewers

Reviewer #1 (Remarks to the Author):

The Editor's question is:

“During the last round of review, one of the other reviewers (Reviewer 3) had some technical concerns about the method used by the authors and this has impact on the validity of the method. We are hoping that you might be willing to assist us further by taking a look at Reviewer 3's comments on the manuscript and the authors response to them. We would be interested in the extent to which Reviewer 3's position is justified and the extent to which the authors have dealt with their comments. You can see these files by using the link provided below.”

Answer: The relevant part is Q2 of the 3rd referee, which makes questionable the theoretical method used in the manuscript. Question is related to the entanglement between the photoelectron and the hole in the HHG process. The reference PNAS 106, 16556 (2009) is mentioned, as well as a tutorial by Smirnova and Ivanov at arXiv:1304.2413 (also published as a book chapter). A “simple” example is given to show that the method used in the manuscript is questionable/meaningless. The authors, of course, disagree and explain why they think that the referee's argumentation is wrong.

My opinion is that the issue with which both the authors and the referee deal is not connected to the present manuscript. It is about which theoretical approach to HHG by a molecular system should be used. One is the QRS theory developed by the authors starting in 2008 for atoms and applied latter to molecules. The other one is the electron-hole dynamics of the above-mentioned PNAS and arXiv papers (and many more). Both theories are approximative and it is difficult to follow all approximations. Theories are confirmed by comparison with the experimental results. In fact, both mentioned theories were “confirmed” by experiments. But the HHG experiments are complicated because of the macroscopic effects and this “confirmation” is only

approximative.

So, the “fight” whose theory is better should be solved finding a problem where these theories give different results and the experiment will decide that (for the parameters of this experiment) one or the other (or neither of them) is correct.

To conclude, I think that this manuscript should be published. It has both theory and experiment and I do not see any flaws or problems.

Reply: We thank the reviewer for the fair arbitration and the positive recommendation of the publication of our work. The QRS theory has been widely used to study HHG from molecules, including the effect of macroscopic propagation. In this work, with the QRS theory, we have well explained our experimentally observed HHG spectra. Based on this fact, in this manuscript, our goal was to solve the inverse problem. Reviewer 1 is correct. If Reviewer 3 wants to challenge the QRS theory, it is better done by a separate publication by him/her. We believe nothing is wrong in our work. In the meanwhile, in the following pages, we will address the remaining concerns of Reviewer 2.

Reviewer #2 (Remarks to the Author):

Note: We have numbered each paragraph from Reviewer 2 with header **Q1**, **Q2**, ... for easier identifications.

Q1: Any publications in professional scientific journals should be written in a rigorous manner, particularly about definition of the phenomenon of interest. The central interest of this article is probing charge migration and a data interpretation method using ML for extracting electron migration dynamic information from HHS data. Electronic wavepacket based on multi-electron correlation probed by HHS specifically regards electron dynamics associated with the recombination process, which is a strong field phenomenon. As the authors also mentioned, the electron dynamics probed by HHS are under the influence of the strong field (laser field). The concept of “charge migration” in Cederbaum’s work is field-free superposition of electronic states due to electron correlation, and therefore is widely cited in statements of its ramifications in chemical reactions and biological systems where electron correlations occur in absence of strong fields. A signature character of the “migration” introduced by Cederbaum et al., which inspired numerous theoretical and experimental efforts even up till today is the revival of the electron density distribution. Not all electronic wavepackets are due to electron correlations and hence lead to “migration” by its origin. Being bound to strong field and limited in probing time range are well known limitations of HHS technique. Simple experimental setup of HHS is associated with challenge of data interpretation which heavily relies on theoretical work. I don’t mean that any work improving HHS is of little value because of the intrinsic drawbacks of the technique. Authors’ claim that the ML-based HHG interpretation method described in this article establishes a general reconstruction scheme for studying ultrafast charge migration is vague and overstating.

Cederbaum’s work was considered by many (see citations) as original work that opened up the research avenue of investigating electron correlation induced charge dynamics (charge migration). As the authors explained, they used this key term in a loose way because they prefer not to adopt a new nomenclature. No clarification of this “adaption” in the original draft. In the revised draft, the authors took no effort to address this confusion due to the shifted meaning or question which would be likely raised by other readers who are familiar with the history of work regarding charge

migration.

Reply: First we thank the reviewer for elaborating the questions above which we had not satisfactorily responded in our previous replies. We apologize for this oversight. To address the questions raised by the reviewer above, let us first focus on our misuse of “charge migration” in our manuscript.

The reviewer stressed that we need to limit the use of “charge migration” to the one used in the original work of Cederbaum and coworkers which are under the field-free condition, and not in our present work using HHS where charge migration occurs in the presence of the driving laser. This distinction was not made clearly in our previous manuscript. We agree with this criticism. In our revised manuscript we have designated the “charge migration” derived from HHG as “laser-driven charge migration”. This change is consistent with the usage in the earlier paper of Kraus et al. [Science 350, 790 (2015)]. Indeed, the main emphasis in our work is to correctly extracting the electron wave packet from the HHG spectra. Such a step is difficult but essential to obtain correct “laser-driven charge migration”. We have no problem to adopt this more precise description.

The reviewer has emphasized that the “charge migration” work of Cederbaum et al. has inspired a lot of recent theoretical and experimental works even up till today. We agree with you in general. But there are a number of limitations in their work.

First, in their model, they consider the evolution of the cation if an electron is suddenly removed from a neutral molecule so they can isolate the charge migration of the cation. Using such a model, they can neglect the coupling, or the entanglement between the escaping electron and the cation. A correct description under general experimental consideration would require the use of density matrix theory, which would deem “charge migration” imprecise. Thus, charge migration and cation electron density has to be understood under this model.

Second, within their model, the correct description of the cation is a complex-valued many-electron wavefunction, or electron wave packet (EWP). This

EWP can be calculated theoretically and the modulus square of the EWP will give the time evolution of the many-electron densities. If the cation can be viewed as a hole (if the molecule is closed-shell), then the time-evolution of the hole density can be viewed as a “movie”. We are aware of many theoretical calculations for such “movies” but none has been “derived” or “extracted” directly from experimental data. These issues have been addressed in the literature, see Stephen R. Leone et al., Nat. Photon. 8, 162-166 (2014) and F. Lepine et al., Nat. Photon. 8, 195-204 (2014). To extract the EWP defined in Cederbaum et al., another probe pulse will have to be used, but such experiments can provide some partial information only, which cannot be used to construct the EWP.

Third, in view of the lack of experimentally derived “charge migration” from pump-probe measurements, it has been realized that HHS provides a pseudo pump-probe scheme. Contrarily to the concern of the reviewer, experimental HHG spectra for the well-studied N_2 and CO_2 molecules have been nicely reproduced based on the QRS model under various laser conditions, including the propagation effect. As described in our manuscript, from HHG, under the equivalent assumption of Cederbaum et al, we can extract the electron dynamics of the cation from the harmonic spectra, except that the electron dynamic evolves in the presence of the laser field. In this manuscript, we have shown the time evolution of the hole dynamics in the coordinate space where the hole density evolves in time. We can generate a “movie” of the movement of the hole density. This is the first accurate construction of laser-driven hole migration using experimental data.

In revising the manuscript, we felt that we need to rewrite the first three paragraphs to make the main motivation and our accomplishment more clearly. We do agree that the examples of N_2 and CO_2 studied here cannot be extended to all other molecules, but in view that there has been no charge migration constructed from the experimental data, the examples presented here definitely can show how laser-driven charge migrations differ in these two molecules. We expect that it can be extended to a few

other small molecules (like benzene) if their HHG spectra have been well studied.

In summary, to respond to Q1, we have designated the “charge migration” derived from HHG as “laser-driven charge migration” in the revised version. In order to show that “laser-driven charge migration” is the best one can get using experimental data, we have rewritten the first three paragraphs completely. We will discuss additional changes after we have answered other questions raised by the reviewer.

Q2: There are other terms that the authors seemingly self-defined and used in a “casual” way, for example, using attosecond photoemission for streaking technique, photo-fragmentation spectroscopy for mass/ion spectroscopy.

Reply: Following the reviewer’s comment, the terms “attosecond photoemission” and “photo-fragmentation spectroscopy” have been changed to “attosecond streaking technique” and “mass/ion spectroscopy” in our revised manuscript.

Q3: I cannot agree with the authors’ statement that “there are no other ways beyond HHG spectroscopy to probe electron dynamics at their natural attoseconds timescale.” RABBIT and energy/angular streaking are common experimental techniques for probing sub fs dynamics, each with their own merits and limits. Isolated pulses of several tens of as duration can be generated via HHG now. Energy streaking directly measures electron energy and when combined with reaction microscope, can do angular dependent study in molecular frame without the need of molecular alignment.

Reply: The above paragraph is from the response of Reviewer 2 to our response to his/her first report. In our revised version we do not make such comments. We agree with the reviewer that the referred experimental techniques can probe electron dynamics at attoseconds timescale. We should have said that these methods would not allow the reconstruction of electron charge density or “charge migration”.

Q4: The authors' response to comment (4) is off the track. They argued that “We had stated qualitatively in the main text (see Lines 8-11 in the 3rd paragraph of the Introduction) and demonstrated quantitatively in Supplementary Fig. 3 that the tools we introduced in this paper is needed for the accurate retrieval of charge migration from HHG spectra.” First, there is no such a section “Introduction” in the manuscript. Second, it is not clear what the authors meant by “stated qualitatively;” words are words. Thirdly, Fig. 3 in the supplement materials depicts dipole moments. What is missing (point in comment 4) is how does it translate into illustrating the “migration” dynamics (evolving spatial distribution). The long argument responding to a referee without making any changes in the manuscript simply doesn't improve the clarity and readability of the article. And the writing of this article in these aspects is far from perfection.

Reply: First, we apologize for our oversight of the definition of the “Introduction” section in our manuscript. In the revised version, we have not those sentences since the first three paragraphs have been rewritten.

The reviewer's criticism is justified. In Supplementary Fig. 3, we have shown that angle-averaged harmonic dipole, especially the phase, deviates distinctly from the single-molecule result. We use this figure to “argue” that one needs to extract the single-molecule dipole in order to obtain accurate “migration” dynamics (the evolving electron density distribution). Indeed, it would be more direct and convincing if we can show that the density distribution from the two methods. We have gladly followed the reviewer's suggestion. As an example, we have plotted the result of N_2 at the alignment angle of 0 degree in Fig. R1. Figure R1(a) shows some selected snapshots of the hole densities reconstructed with the retrieved single-molecule dipoles. Figure R1(b) is the corresponding results reconstructed with the angle-averaged dipoles. For comparison, we have also plotted the TDDFT results in Fig. R1(c). It's evident that the reconstructions with the angle-averaged dipoles are very different from that with the single-molecule dipoles and show worse agreement with the TDDFT results. This

is mainly due to the underestimated contribution of the HOMO-1 orbital of N_2 molecule in the angle-averaged dipoles. This result indicates a severe influence of the angular averaging on the accuracy of the reconstruction. We have added this result to the revised Supplementary (see Supplementary Note 6) to clarify this point. We thank the referee for making this suggestion.

Fig. R1: Snapshots of the reconstructed hole densities in N_2^+ for the alignment angle of 0 degree. (a) is the results reconstructed with the retrieved single-molecule dipole moments. (b) same as (a), but for the reconstruction with the angle-averaged harmonic dipole moments in our experiment. (c) plots the TDDFT results for comparison.

Q5: This work has solid results and shows a new method. It is expected that the new method can improve HHS data interpretation which will reveal interesting information of certain processes that cannot be accessed with conventional HHS data analysis. It is not convincing that it will have a strong general impact in the corresponding field, though the topic this work concerns is of intense interest. It is certainly worth publication, but not in Nature Communications, in my opinion. The manuscript needs to be improved as pointed out above for publication.

Reply: We thank the reviewer for agreeing with us that this work is of intense interest and scientifically sound. On the other hand, we respectfully disagree with the reviewer's recommendation that this work is not worthy of publication in Nature

Communications. We hope that the reviewer will change your recommendation after reading our revised version.

We wish to say that the points raised by the reviewer in the two rounds of reports are mostly about our misuse of “charge migration” for describing the electron dynamics in the presence of the laser field. Other questions are related to the lack of clarity in some sentences or additional figures to help strengthen our results. We regret to point out some of them are due to the limitation of our language skill. With your help we hope that all of these questions have been resolved in the revised version.

Before reading our revised version, the reviewer stated above that “*the new method can improve HHS data interpretation which will reveal interesting information of certain processes that cannot be accessed with conventional HHS data analysis. It is not convincing that it will have a strong general impact in the corresponding field*”. Possibly because it was not well addressed in our previous manuscript, the reviewer may have missed the major achievement of our work. We emphasize that our work is **the first time** that attosecond “electron dynamics of the cation” has been extracted **correctly** from the harmonic spectra. In almost all previous studies on HHS, the single-molecule harmonic amplitude and phase have been directly taken from the experimental data. Such a practice is intrinsically incorrect and would introduce large errors in the reconstructions (as shown in Fig. R1 above). Without the processing of the experimental HHG data as presented in our work, the extracted single-molecule parameter like molecular orbital from tomography experiment at most only has qualitative significance, see Nature 432, 867 (2004) by Itatani et al., “Tomographic imaging of molecular orbitals”. Our work solves the long-standing challenge in HHS. The extracted electron dynamics in our work is definitely much more useful than the extraction of molecular orbital of N₂. Molecular orbital of N₂ is well known, but electron dynamics in the laser field is not. The density plots or movies presented in our results show how they differ in N₂ and CO₂. They have never been seen until this work is published. Granted, this method cannot be applied to any molecules, but not

all of the new experimental or theoretical methods can be applied to all molecules, especially when the complexity increases. For example, photoelectron spectra are powerful research tools for small molecules. Few experiments would study the photoelectron momentum spectra of a large molecule. The present work definitely can be extended to a number of other molecules to investigate electron dynamics. The list is not limited to the present two molecules. Moreover, our work is not just for charge migration, but also for any ultrafast measurement with HHG if single-molecule information is to be extracted. We believe our work is a big step forward beyond what was reported on HHS previously and will have an important impact in the corresponding field.

We also want to call attention to the reviewer that we have rewritten the conclusion and the paragraph before the conclusion and add a new Supplementary Note 7. In the latter, we discuss the field-free charge migration in these two molecules. The last two figures in Supplementary Note 7 compare how the migration differs when the field is present or not, for different orientations of the molecules. The conclusion has been rewritten, removing all of the more speculative sentences, and pointing out the present retrieval methods can be and should be extended to other ultrafast phenomena if one wants to extract single molecule parameter using gas targets. All the revisions are marked by red color in the revised version.

We hope that the reviewer will support the publication of this revised manuscript after reading our responses and our clarifications to your questions. We thank you again for your time and effort in helping us to improve this manuscript and that you will be proud to see that this work appears in Nature Communications.

Reviewer #3 (Remarks to the Author):

First of all, I apologize to the authors, who seem to have taken my original report, which was intended strictly as a opinion on the `_physics_`, as a `_personal_` attack. Nothing could be further from the truth, and I deeply regret that my completely frank and admittedly robust opinion - something which I believe is the point of anonymous refereeing - seems to have caused the authors such a personal offence. Once again, I do sincerely apologize.

Returning to the physics: To my great disappointment, the revised manuscript does not address any of the major issues brought up in my original report. Upon carefully reading the authors' rebuttal, I remain convinced that my original arguments are completely valid. I am most grateful to the authors for pointing out the obvious typing mistakes in some of the formulae I gave: it is of course true that the coordinate index should be changed in one of the functions. Typing TeX "blind", without rendering the formulae is unfortunately error-prone.

While the suggestion by the authors that I should recommend publications of the results I believe to be wrong, alongside the explanation of why, is certainly enticing, I must respectfully decline the offer. It would not be fair to make public comments anonymously - and the authors' rebuttal seems to imply that the authors do intend to make it a personal fight, rather than a discussion of physics.

Upon careful reconsideration, my original assessment and recommendation therefore stand.

Reply: First, we thank Reviewer 3 for providing the detailed report on our work in the first round of review. Some of the questions raised have helped us to improve the presentation. However, we have not expected that the reviewer considered our rebuttal as a personal fight, rather than a discussion of physics. We have never intended to make the rebuttal a personal fight. In fact, we have made detailed point-to-point responses to each question of the reviewer and tried our best to explain why we

disagree with the reviewer. All our replies are sound and discussed based on the physics. We firmly believe that our work has been built on solid foundation and nothing is wrong in the results.

REVIEWER COMMENTS

Reviewer #2 (Remarks to the Author):

Thank the authors for their effort in improving the manuscript in scientific rigorousness and clarifying their achievement in this work and the importance of it in their view.

The authors cited works in Refs [15,16] and stated that “It (field-free charge migration) cannot be directly obtained without applying a probe pulse at different time delay which inevitably would modify the field-free charge migration. Thus, charge migration was studied almost exclusively based on theoretical calculations only [1, 2, 17–19]. It is still desirable to obtain charge migration directly from experiment.” and then mentioned HHG spectroscopy as “an alternative.”

This is misleading. The requirement of the presence of a strong laser field in HHG spectroscopy is exactly the root of the limitation of HHG spectroscopy, which lies in not the technical merit but what the method can be applied to study. HHG spectroscopy limits to probing strong-field phenomena, rescattering phenomena, specifically. “(Strong) Field-free charge migration” is the ubiquitous process in natural systems that many research groups work on investigating, albeit the challenge of probing the dynamics as the authors mentioned. This is not a limitation of someone’s work, but a challenging topic with broader impacts.

This “misunderstanding” by the authors dictates the presentation of the entire article (and their response), making their work appear as an improved technique of a narrower influence. The authors seemingly think the discussion is simply about a terminology that they didn’t pay enough attention to use properly. It is still great work if the presented method improves substantially the studies of strong-field phenomena. However, it is now clearly shown in the current manuscript.

The angle dependence of the dipole moment is intrinsic in HHS spectroscopy. Since molecular alignment won’t be 100% efficient, indeed ideally, this angle dependence should be deconvolved to extract accurate dipole moments and therefore accurately extract information of the electron dynamics. This solution is not an idea one can hardly think of. The core merit of this work is machine learning is applied to achieve deconvolution of the angular distribution and the authors show the results using this method. To evaluate the importance of this work, a key question would be how much a difference this method makes in experiments feasible nowadays. No quantitative data about this are presented in the main body of the manuscript and only Fig 12 in the supplement is shown for N₂ with an alignment degree of $\langle \cos^2\theta \rangle = 53$ (not great alignment; roughly twenty years ago, it was shown that N₂ can be aligned with $\langle \cos^2\theta \rangle = 65$ (PHYSICAL REVIEW A 68, 023406, 2003)). There is no presentation of how the new method improves the accuracy of the retrieved dynamics information in a different molecule that is less easy to

align, CO₂⁺. Even if the focus of this work is about the observed electron dynamics as shown in several figures in the main text, an insightful explanation of these dynamics is expected but is missing in the current manuscript. The authors only state that a swirl motion of the electron was revealed by the new method.

This work shows solid results from both experiments and theoretical calculations and some new discovery. No doubt, it is worth publication. However, as explained above, I won't recommend publication in Nature Communications.

Response to Reviewer comments

Reviewer #2 (Remarks to the Author):

Note: We have numbered each question from Reviewer 2 with header **Q1**, **Q2**, ... for easier identifications.

Q1: Thank the authors for their effort in improving the manuscript in scientific rigorousness and clarifying their achievement in this work and the importance of it in their view.

The authors cited works in Refs [15,16] and stated that “It (field-free charge migration) cannot be directly obtained without applying a probe pulse at different time delay which inevitably would modify the field-free charge migration. Thus, charge migration was studied almost exclusively based on theoretical calculations only [1, 2, 17–19]. It is still desirable to obtain charge migration directly from experiment.” and then mentioned HHG spectroscopy as “an alternative.”

This is misleading. The requirement of the presence of a strong laser field in HHG spectroscopy is exactly the root of the limitation of HHG spectroscopy, which lies in not the technical merit but what the method can be applied to study. HHG spectroscopy limits to probing strong-field phenomena, rescattering phenomena, specifically. “(Strong) Field-free charge migration” is the ubiquitous process in natural systems that many research groups work on investigating, albeit the challenge of probing the dynamics as the authors mentioned. This is not a limitation of someone’s work, but a challenging topic with broader impacts.

This “misunderstanding” by the authors dictates the presentation of the entire article (and their response), making their work appear as an improved technique of a narrower influence. The authors seemingly think the discussion is simply about a terminology that they didn’t pay enough attention to use properly. It is still great work if the presented method improves substantially the studies of strong-field phenomena. However, it is not clearly shown in the current manuscript.

Our extended response to Q1

(Note: words in **red** are from the referee)

Reply: The above three paragraphs finally made it clear to us why this referee has repeatedly turned down our manuscript despite that we have addressed all of the questions raised by him/her. The two key troublesome but clear sentences by the referee in this report are

- (i) “(Strong) Field-free charge migration” is the ubiquitous process in natural systems that many research groups work on investigating, albeit the challenge of probing the dynamics as the authors mentioned.
- (ii) “... their work appears as an improved technique of a narrower influence”.

We will elaborate why his/her understanding is wrong and his/her opinion is biased in the following. Please take note that our rebuttal meant serious pure scientific argument. No personal attack is intended.

First, the word “(strong)” in (i) is misleading since if it is field free, what is “strong”? In (ii) “narrow influence” by the referee means that strong field HHG spectroscopy is a narrow topic, playing little role in the “ubiquitous field-free charge migration”.

As the authors, we are at a disadvantage to criticize the referee for his/her lack of general knowledge on the topic of our paper, thus leading to his/her prejudice against using strong field probe for electron dynamics. For clarity, below we expose these shortcomings separately using points **P1** to **P6**.

P1. Owing to his/her dislike of strong field probing of electron dynamics, the referee has decided from the beginning to turn down our manuscript for publication in Nature Communications. If this is the reason, why he/she did not say so clearly from the start but continued to ask us questions on the “technical” details of our manuscript? He/She has forced us to answer many questions and knowing that even if we addressed all his/her questions well, he/she still would not change his/her verdict. Even in this newest round, in the 2nd half of the report he/she still is not shy to ask us more questions on strong field physics again. By asking us more questions without intending to possibly accept our manuscript has resulted in holding this manuscript as a hostage.

In short, the referee’s repeated questions on details of HHG spectroscopy while deeply detesting it have confused us in our prior dialogs with him/her.

P2. The referee is a strong believer of “field-free charge migration” that was first coined by Cederbaum and coauthors (Refs. [1,2] in our manuscript), saying that it is the “ubiquitous process in natural systems”. Field-free charge migration defined in the original papers is the time-dependent density of the hole after an electron is suddenly removed from a closed-shell molecule. In elementary quantum mechanics, such a sudden removal is to create a **wave packet** which is a superposition of wave functions

of eigenstates. This wave packet is a complex-valued wave function. The modulus square of this wave packet is the charge density and its time dependence has been called charge migration [1,2]. Since charge density is positive definite, one can plot it as a function of time to make an “electron” or “hole” movie. Such “movies” can be easily constructed from theoretical calculations.

The referee is wrong to say that “field-free charge migration” is ubiquitous in natural systems. The correct and ubiquitous one is the complex-valued wave packet. By taking the modulus square of the wave packet, the phase information is lost and the description of electron dynamics would be completely wrong. The only advantage of presenting charge density is that one can display its time-dependence as a “movie”. In most quantum chemistry calculations, the wave packet is calculated and then the charge density is obtained. Similarly, charge density cannot be directly obtained from the experimental data. For example, in Eq. (14) of our manuscript, it is the wave packet that is retrieved, and the charge density is obtained by taking its modulus square.

In any experiment, a measurement would involve the wave function or wave packet, not the charge density. Thus, the field-free charge migration proposed in Cederbaum et al. cannot be determined directly from experimental data. In other words, whether it is a scattering calculation or a retrieval of electron dynamics from the experimental data, it is the wave packet that is providing the correct information, not the charge density.

Charge density plots are useful only for making a “movie” to provide a partial “understanding” of electron dynamics. Its role is limited in quantum mechanics. In classical physics a “movie” may be considered to be “real”, but not in quantum mechanics.

In short, the referee has overly emphasized the significance of time-dependent charge density for describing electron dynamics. According to quantum mechanics, it should be the time-dependent wave packet.

P3. In his/her point (i), the referee noted that “many research groups work on investigating, albeit the challenge of probing the dynamics as the authors mentioned”. Based on the discussion in **P2**, it is not conceivable that any (instead of “many”) research group(s) can study charge migration directly. In the definition of charge migration of the original authors, there has never been a proposal on how to “measure” charge migration. The referee stated that many research groups are working on it. At the best, they can work on retrieving the wave packet, not the charge density. Retrieving the wave packet has additional challenges since one also has to extract the phase from the experiment. For example, in

our paper, we need to extract the amplitude and phase of the C-coefficients which are shown in Figs. 2 (d, g, h) in the main text.

In short, as shown in our manuscript, the complex wave packet should be extracted to describe electron dynamics, not the charge migration or charge density.

P4. The referee disapproved HHG spectroscopy because a strong field is used. Clearly, a weak field is preferred if one wants to probe electron dynamics. The real question is whether a weak field can be used to probe electron dynamics at attoseconds timescale? An attosecond probe pulse in the linear regime won't provide enough fluence to generate measurable signals for detection. Comparing the fluence of a typical 100 ps light from synchrotron radiation to a 100 as XUV pulse, just from the duration alone, the fluence of an attosecond pulse is down by a factor of one million. To probe ultrafast phenomena, no doubt intense light sources are needed. Thus, intense infrared lasers and accelerator-based free electron laser facilities like LCLS and other similar kinds in the world have been built in the past decade. If strong fields are not needed why people would spend billions or hundred of millions to build such large facilities.

Another key issue is that a linear probe in general cannot probe the phase. Previously, in a paper by Zhao et al, Phys. Rev. A 98, 053404 (2018), the authors theoretically considered the reconstruction of the electron wave packet from the energy and angular photoelectron spectra for one-photon ionization of a one-electron model argon atom by an attosecond XUV pulse. Even for this simplest problem, it was found that it is not possible to retrieve the whole wave packet using the full energy and angular distributions of the photoelectrons alone. An extra energy dependent phase is needed for complete reconstruction. The latter is obtained by using an XUV+IR two-color pulses with varying time delay, illustrating that a nonlinear probe is needed. As far as we know, a nonlinear probe is required if the wave packet is to be retrieved.

In our previously resubmitted revised manuscript, we have made an objective statement about this practical problem in probing attosecond charge migration. But the referee thinks it is misleading. The referee refuted us by saying that **"This is not a limitation of someone's work, but a challenging topic with broader impacts"**. Our explanation above concludes that charge migration or time-dependent charge density is limited for describing electron dynamics. Charge migration may have a broad "interest" because it provides a "movie", but it won't have a broad impact on attosecond electron dynamics.

Moreover, we did not say it is “a limitation of someone’s work”. We have only made a scientific argument on this practical problem.

In short, referee’s despire on strong field for probing attosecond electron dynamics is unwarranted.

P5. The use of HHG spectroscopy for probing electron dynamics has been discussed widely in the literature, and has been featured prominently, for example, in a Focus/Review Article in **Nature Photonics** [Ref.16 in our manuscript], by F. Lepine, M. Ivanov and M. Vrakking. Specifically, on pages 197-198 of Ref.16, under the heading of “First experimental protocol: attosecond measurements without attosecond pulses”. Meanwhile, a similar Focus/Review Article in **Nature Photonics** [Ref.15 in our manuscript], by Stephen Leone and about 20 additional leading attosecond scientists, also has addressed this issue. The basic idea for extracting electron dynamics from HHG spectra is easy to understand, but to extract it from experimental data is much more difficult, as addressed in the Introduction of our manuscript. The first such experiment using HHG spectroscopy is in the paper by Kraus et al, published in **Science** [Ref.28]. In our manuscript, using machine learning and two-color probe pulses we have removed key assumptions made in this early paper so we obtained electron dynamics of a single molecule from experimental harmonic spectra, as explained in the Introduction of our manuscript. The referee bluntly disputed our use of HHG spectroscopy because it uses strong fields also discredited prior researches carried out by hundreds of scientists.

In short, referee’s objection to HHG spectroscopy technique is unfounded.

P6. Repeating the last five points for the referee, first, the “ubiquitous field-free charge migration” defined by Cederbaum et al. has never addressed the measurement issue. To study electron dynamics, the electron wave packet itself is the one to obtain. Charge density or charge migration can be considered to be a compromise to assist visualization (for whatever its worth) after the wave packet has been obtained. We put the wave packet at the higher level, while the referee thinks charge density is at that level. This is incorrect.

The referee disapproved using strong field to probe electron dynamics. This is generally understood since nonlinear interaction is much harder to calculate or understand. However, this preference has been checked against experimental feasibility. To obtain measurable signals a certain amount of impulse acting on the electrons is needed. Due to the short duration of an attosecond pulse, without a strong field, the fluence is too small to probe. In the meanwhile, to extract the phase of the wave packet,

nonlinear interaction is generally needed. We remind the referee that almost all the experiments using attosecond pulses are performed using the two-color XUV+IR pulse.

In short, we claimed that referee's negative recommendation on our manuscript is not justified since it is based on his/her misunderstanding of the topic we have presented.

Epilogue

Coming back to the core issue of our manuscript, we demonstrated how we can construct the electron (or hole) wave packet from which charge density can be presented as a "movie". The HHG spectroscopy is used where the initial wave packet was created by ionization of the molecule and the probe is the electron recombination with the emission of harmonics. For years we have demonstrated that harmonic generation can be described using our QRS theory. According to this model, the first step is the nonlinear tunnel ionization, the recombination step is linear, since the latter is an inverse of conventional photoionization. As discussed in the Methods section of the main text, the QRS model allows us to extract the wave packet of the hole dynamics. To retrieve single-molecule dynamics using experimental harmonic spectra, additional effort is needed which has been documented in our manuscript.

We emphasize that HHG spectroscopy is a well-known and widely used technique in the field of attosecond science and has a large research community. Our work solves the long-standing deficiency in HHG spectroscopy. It is **the first time** that **angle-resolved** attosecond electron dynamics in molecules has been extracted **correctly** from the harmonic spectra. As far as we know, we have **the only result** for charge migration for single molecules, even though the electron is under the laser field. We strongly disagreed with the referee's statement: **"The requirement of the presence of a strong laser field in HHG spectroscopy is exactly the root of the limitation of HHG spectroscopy"**. Indeed, without strong fields, there would be no attosecond science today. The importance of this field is also reflected in the many papers on HHG that had been published in Nature and its sister journals, as well as in Science. For example, J. Itatani et al., **Nature** 432, 867 (2004), T. Kanai et al., **Nature** 435, 470 (2005), S. Baker et al., **Science** 312, 424 (2006), B. McFarland et al., **Science** 322, 1232-1235 (2008), O. Smirnova et al., **Nature** 460, 972 (2009), S. Haessler et al., **Nat. Phys.** 6, 200 (2010), D. Shafir et al., **Nat. Phys.** 5, 412 (2009), H. J. Wörner et al., **Science** 334, 208–212 (2011), P. M. Kraus et al., **Science** 350, 790 (2015), the recent review P. Peng et al., **Nature Review Physics** 1, 144–155 (2019), and so on.

We feel that we have exposed the shortcomings of the referee's understanding of probing attosecond electron dynamics in our lengthy presentation. If the referee still disagrees with our rebuttal, we suggest that a fair verdict to the outcome of this manuscript is for you to recommend its publication in Nature Communications, with the condition that our questions-and-answers from the review processes be published in parallel in the COMMENT section of the published article. When a strong dispute exists between the authors and the referee on such an important problem, sharing such arguments with the readers should be encouraged. After all, we are arguing on science. Finally, we still thank the referee for challenging us to raise these issues presented above.

The referee has asked some new questions in the following paragraph that were not asked before. They are all related to strong field physics.

Q2: The angle dependence of the dipole moment is intrinsic in HHS spectroscopy. Since molecular alignment won't be 100% efficient, indeed ideally, this angle dependence should be deconvolved to extract accurate dipole moments and therefore accurately extract information of the electron dynamics. This solution is not an idea one can hardly think of. The core merit of this work is machine learning is applied to achieve deconvolution of the angular distribution and the authors show the results using this method. To evaluate the importance of this work, a key question would be how much a difference this method makes in experiments feasible nowadays. No quantitative data about this are presented in the main body of the manuscript and only Fig 12 in the supplement is shown for N₂ with an alignment degree of $\langle \cos^2\theta \rangle = 0.53$ (not great alignment; roughly twenty years ago, it was shown that N₂ can be aligned with $\langle \cos^2\theta \rangle = 0.65$ (PHYSICAL REVIEW A 68, 023406, 2003)). There is no presentation of how the new method improves the accuracy of the retrieved dynamics information in a different molecule that is less easy to align, CO₂⁺. Even if the focus of this work is about the observed electron dynamics as shown in several figures in the main text, an insightful explanation of these dynamics is expected but is missing in the current manuscript. The authors only state that a swirl motion of the electron was revealed by the new method.

Reply: We address each additional criticism in separate paragraphs for clarity.

First, in our last revision, we have already stressed the influence of alignment-averaged harmonic spectra on the reconstruction of charge migration (Supplementary Note 6). In this report, the referee accused us that no quantitative data on this issue are presented in the main text. This criticism is

inappropriate. For the readability of the manuscript, we would only put the most relevant results in the main text. Other less relevant data belong to the supplement which is an integral part of this manuscript.

Second, the referee complained that $\langle \cos^2\theta \rangle = 0.53$ in our work is not a great alignment and pointed out a better alignment of 0.65 in PRA 68, 023406, 2003. This remark has no significance. First, in the PRA paper, the angle ϑ is between the alignment axis and the projection of the three-dimensional fragment velocity onto the polarization plane, which is different from the commonly defined azimuthal angle θ in spherical coordinate system (This has been clarified in the first paragraph in the right column on page 4 of the PRA paper). For an isotropic distribution, $\langle \cos^2\theta \rangle = 1/3$, while $\langle \cos^2\vartheta \rangle = 0.5$. In the PRA work, it measured $\langle \cos^2\vartheta \rangle = 0.65$, not $\langle \cos^2\theta \rangle = 0.65$.

We should mention that $\langle \cos^2\theta \rangle = 0.53$ is a typical level for most molecular alignment experiments, (e.g. in the science paper by Kraus et al. [Ref. 28], $\langle \cos^2\theta \rangle = 0.5 \sim 0.55$). For a slightly higher alignment degree, say $\langle \cos^2\theta \rangle = 0.65$, the averaged complex harmonic dipole (especially the phase) still deviates severely from the single-molecule dipole (as shown in Supplementary Fig. 3), and it would not improve the accuracy of the reconstruction much. To demonstrate this point, we have reconstructed the migration dynamics of N_2 with $\langle \cos^2\theta \rangle = 0.65$. As shown in Fig. R1(c), the result is close to that reconstructed with $\langle \cos^2\theta \rangle = 0.53$ [Fig. R1(b)], and shows large difference from the single-molecule result [Fig. R1(a)] as well as the TDDFT simulations [Fig. R1(d)]. We also remind the referee that the influence of angular average to the complex harmonic dipole is already shown in Supplementary Fig. 3 for many different values of $\langle \cos^2\theta \rangle$. It is meaningless and impractical to show the retrieved charge migration movies for all the $\langle \cos^2\theta \rangle$.

Upon referee's request, we have performed similar reconstructions for CO_2^+ . The results are shown in Fig. R2. One can clearly see that the hole dynamics reconstructed with the angle-averaged dipoles [Fig. R2(b)] differs from that with the single-molecule dipoles [Fig. R2(a)] and shows worse agreement with the TDDFT results [Fig. R2(c)]. We have added the results of CO_2^+ to the supplement (see Supplementary Note 6).

Finally, the referee complained that "an insightful explanation of these dynamics is expected but is missing in the current manuscript". We think the referee should know better that charge density is the modulus square of complex-valued wave packet. The wave packet is the interference of the complex wave functions. There is no chance to interpret it without the calculation. In the meanwhile, charge density is actually a probability density of the hole which is not directly governed by an equation of

motion. Charge migration was used only if one wants to make a “movie”. Note that the charge migration also depends critically on the alignment angle of the molecule. Only real calculations can show what they are.

Fig. R1: Snapshots of the reconstructed hole densities in N_2^+ for the alignment angle of 0 degree. (a) is the results reconstructed with the retrieved single-molecule dipole moments. (b) same as (a), but for the reconstruction with the angle-averaged harmonic dipole moments in our experiment ($\langle \cos^2\theta \rangle = 0.53$). (c) same as (b), but for the alignment degree of $\langle \cos^2\theta \rangle = 0.65$. (d) plots of TDDFT results for comparison.

Fig. R2: Snapshots of the reconstructed hole densities in CO_2^+ for the alignment angle of 0 degree. (a) is the results reconstructed with the retrieved single-molecule dipole moments. (b) same as (a), but for the reconstruction with the angle-averaged harmonic dipole moments in our experiment. (c) plots the TDDFT results for comparison.

Q3: This work shows solid results from both experiments and theoretical calculations and some new discovery. No doubt, it is worth publication. However, as explained above, I won't recommend publication in Nature Communications.

Reply: Like the previous two reports, the referee continued to say that our work has solid results and new discovery, but turned down our work again without giving any solid reason. In the first part of this report, finally we realized that the referee is strongly against the HHG spectroscopy because of his/her failure to understand that strong field plays an essential role on dynamics at attoseconds timescale. We have exposed his/her misconception thoroughly in six points in the first part of our response.

All the new questions have been addressed in the literature and they are not essential for the understanding of our manuscript.

Finally, we repeat the last paragraph presented earlier:

We feel that we have exposed the shortcomings of referee's understanding of probing attosecond electron dynamics in our lengthy presentation. If the referee still disagrees with our rebuttal, we suggest that a fair verdict to the outcome of this manuscript is for you to recommend its publication in Nature Communications, with the condition that our questions-and-answers from the review processes be published in parallel in the COMMENT section of the published article. When a strong dispute exists between the authors and the referee on such an important problem, sharing such arguments with the readers should be encouraged. After all, we are arguing on science. Finally, we still thank the referee for challenging us to raise these issues presented above.

Revisions to the new submitted manuscript

To accommodate the minor corrections from the referee in his/her latest report, we document the minor corrections and addition made:

(1) To avoid the endless dispute in Q1 of the referee, we have removed the sentences mentioned by the referee in the revised manuscript.

(2) We have added a brief discussion about the influence of alignment degree on the reconstruction in Supplementary Note 6. Meanwhile, the results of CO_2^+ in Fig. R2 have also been added to the supplement (see the new Supplementary Figure 13).

REVIEWER COMMENTS

Reviewer #2 (Remarks to the Author):

Thank the authors for their effort to address their views, provide explanations and express disagreements. I think their strong belief that their manuscript meets all requirements for publication in Nature Communications prevents them from grasping a good understanding of my points about the reasons I won't recommend publication. I doubt the authors understand the purpose of the review process and their judgement and perspective "Owing to his/her dislike of strong field probing of electron dynamics, the referee has decided from the beginning to turn down our manuscript ... He/She has forced us to answer many questions and knowing that even if we addressed all his/her questions well, he/she still would not change his/her verdict." is absurd.

The authors' rebuttal focuses on repeating the merits of this work and disagreements which remain unresolved. Unfortunately, this kind of effort doesn't improve the manuscript and doesn't change my judgment. With no more questions, I don't recommend publication of this manuscript in Nature Communications. The authors should address to the editor their concerns about the qualification of a referee and send to the editor the request for publication of referees' reports and their response as the authors would like to.

Reviewer #4 (Remarks to the Author):

Review report on NCOMMS-21-31786C

The authors report the filming movies of attosecond charge migration in single molecules from the high harmonic spectra in two-color laser pulses assisted by the machine learning algorithms.

It is noteworthy that they could construct the movie of "laser-driven charge migration" after tunneling ionization of the N₂ and CO₂ molecules with the time interval of 50 attoseconds. Even though the extension of the presented methods to the more complicated or nonlinear molecules and/or molecules having dense electronic states, but the presented results are one of the steps for investigating the details on the intramolecular charge transfer processed driven by the electron correlation.

Therefore, I can strongly recommend the paper to be published in the Nature Communications.

The dynamics measured in this work is “laser-driven charge migration” as clearly depicted in their manuscript. Even though the dynamics is different from the “field-free charge migration” processes induced by ultrashort short-wavelength light sources, but the interpretations on the measured charge migration process are still very interesting.

Probably, the authors should briefly discuss the experimental strategy for minimizing or reducing the strong field effect onto the charge migration processes. For example, the high harmonic spectroscopy using the few-cycle laser pulses of fundamental pulse (not necessary to generate few-cycle pulse for the second harmonic) might be one of the solutions.

In addition, the authors should explain why the filmed movies spans only from approximately 1 fs to 1.5 fs for the non-expertized readers on the attosecond science. Without explanation, the non-expertized readers might feel that the filmed movies are quite short.

If the similar experiments are performed with the long wavelength driving laser, the time span of the movie can be expanded by sacrificing the timestep of the movie and it will be able to capture the certain effect of nuclear motions (decoherence and recursion).

Responses to Reviewer Comments

Reviewer #2 (Remarks to the Author):

Comment: Thank the authors for their effort to address their views, provide explanations and express disagreements. I think their strong belief that their manuscript meets all requirements for publication in Nature Communications prevents them from grasping a good understanding of my points about the reasons I won't recommend publication. I doubt the authors understand the purpose of the review process and their judgement and perspective "Owing to his/her dislike of strong field probing of electron dynamics, the referee has decided from the beginning to turn down our manuscript ... He/She has forced us to answer many questions and knowing that even if we addressed all his/her questions well, he/she still would not change his/her verdict." is absurd.

Reply: We thank the referee for reading our manuscript and asking questions at each round of the review process. We apologize for speculating the reason for your negative recommendation in the above cited sentences, but it is true that you have never provided us the reason why our manuscript does not meet the requirements for publication in Nature Communications. A single referee should not decide the fate of a manuscript unless the flaws are clearly documented. Under this circumstance, we think the editor did it correctly to seek report from another referee. Once again, we thank you for your effort.

Comment: The authors' rebuttal focuses on repeating the merits of this work and disagreements which remain unresolved. Unfortunately, this kind of effort doesn't improve the manuscript and doesn't change my judgment. With no more questions, I don't recommend publication of this manuscript in Nature Communications. The authors should address to the editor their concerns about the qualification of a referee and send to the editor the request for publication of referees' reports and their response as the authors would like to.

Reply: With the new strong report from referee 4, together with the two earlier reports from referee 1, we believe that our manuscript meets the requirements for publication in Nature Communications.

Reviewer #4 (Remarks to the Author):

Comment: The authors report the filming movies of attosecond charge migration in single molecules from the high harmonic spectra in two-color laser pulses assisted by the machine learning algorithms.

It is noteworthy that they could construct the movie of “laser-driven charge migration” after tunneling ionization of the N₂ and CO₂ molecules with the time interval of 50 attoseconds. Even though the extension of the presented methods to the more complicated or nonlinear molecules and/or molecules having dense electronic states, but the presented results are one of the steps for investigating the details on the intramolecular charge transfer processed driven by the electron correlation.

Therefore, I can strongly recommend the paper to be published in the Nature Communications.

The dynamics measured in this work is “laser-driven charge migration” as clearly depicted in their manuscript. Even though the dynamics is different from the “field-free charge migration” processes induced by ultrashort short-wavelength light sources, but the interpretations on the measured charge migration process are still very interesting.

Reply: We thank the referee for pointing out the achievement we have made on charge migration presented in our manuscript and the positive recommendation of the publication of our work.

Below we address the questions raised by the referee. We have also revised our manuscript according to referee’s suggestions. We believe that the revised version now can be accepted for publication.

Comment: Probably, the authors should briefly discuss the experimental strategy for minimizing or reducing the strong field effect onto the charge migration processes. For example, the high harmonic spectroscopy using the few-cycle laser pulses of fundamental pulse (not necessary to generate few-cycle pulse for the second harmonic) might be one of the solutions.

Reply: We share with the referee’s desire for such a possibility and agree that using a few-cycle fundamental laser may help to reduce the strong field effect on the charge migration process in the measurement. In reality, driven by a few-cycle laser pulse with the same peak intensity, the range of the generated harmonics will be about the same as by a longer pulse, but each harmonic will be less sharp in the plateau region and become a continuum in the cutoff region, then it may increase the difficulty to map the time to each high harmonic.

Revision: We have added a few sentences at the end of the second paragraph on page 11 (marked in red) in the main text to comment on the use of few-cycle pulses for high harmonic spectroscopy.

Comment: In addition, the authors should explain why the filmed movies spans only from approximately 1 fs to 1.5 fs for the non-expertized readers on the attosecond science. Without explanation, the non-expertized readers might feel that the filmed movies are quite short.

Reply: The 1.0 fs to 1.5 fs limitation is caused by the 800 nm laser used in the experiment.

Revision: We have added the explanation at the end of the first paragraph on page 11 (marked in red).

Comment: If the similar experiments are performed with the long wavelength driving laser, the time span of the movie can be expanded by sacrificing the timestep of the movie and it will be able to capture the certain effect of nuclear motions (decoherence and recursion).

Reply: We agree with the reviewer that if long wavelength driving laser is used, then the time span of the movie will increase (proportional to the wavelength). For example, if one uses 2400 nm laser, the time span will be from 3 fs to 4.5 fs, roughly. The increased time window may be conducive to explore the effect of nuclear motions on the electron dynamics at longer time. However, in experiment with longer wavelength, the harmonic yields will be severely reduced as the yield declines approximately by $\lambda^{-(5-6)}$. Harmonic spectra of N₂ and CO₂ generated with wavelength of about 2000 nm have been reported for unaligned molecules only. For the goal of reconstructing charge migration, experimental studies for aligned molecules by long wavelength lasers should be carried out first.

Revision: We have discussed this point at the end of the first paragraph on page 11 (marked in red).

REVIEWERS' COMMENTS

Reviewer #4 (Remarks to the Author):

The authors corrected their manuscript following the comments raised by the reviewers accordingly.

I can strongly recommend the paper to be published in the Nature Communications in the present form.

Reviewer #4 (Remarks to the Author):

The authors corrected their manuscript following the comments raised by the reviewers accordingly.

I can strongly recommend the paper to be published in the Nature Communications in the present form.

Reply: We thank the referee very much for the positive recommendation.